# Including the Phosphorus cycle into the LPJ-GUESS Dynamic Global Vegetation Model (v4.1, r10994) – Global patterns and temporal trends of N and P primary production limitation.

Mateus Dantas de Paula[1], Matthew Forrest[1], David Warlind[4], João Paulo Darela Filho[2], Katrin Fleischer[5], Anja Rammig[2], Thomas Hickler[1,3]

[1]Senckenberg Biodiversity and Climate Research Centre (SBiK-F), Georg-Voigt-Straße 14-16, 60325, Frankfurt am Main, Germany
[2]Professorship for Land Surface-Atmosphere Interactions, Technical University of Munich, Hans-Carl-v.-Carlowitz-Platz, 2, Freising, 85354, Bavaria, Germany
[3]Department of Physical Geography, Geosciences, Johann Wolfgang Goethe University of Frankfurt, Frankfurt, Germany
[4]Department of Physical Geography and Ecosystem Science, Lund University, Lund, Sweden
[5]Section Systems Ecology, Amsterdam Institute for Life and Environment (A-LIFE), Vrije Universiteit Amsterdam, Netherlands

*Correspondence to*: Mateus Dantas de Paula (mateus.dantas@senckenberg.de) and Thomas Hickler (thomas.hickler@senckenberg.de)

**Abstract.** Phosphorus (P) is a critical macronutrient for plant growth, often limiting plant production in areas where plant demand is higher than soil supply. In contrast to nitrogen (N), P cannot be sourced from the atmosphere, therefore where it is rare, it becomes a strong constraint of primary production. Due to this, most DGVMs are incorporating a prognostic P cycle in addition to N, improving their ability to correctly predict stocks and fluxes of carbon, and how climate change may impact N and/or P limitations to soil processes and plant productivity.

We included the P-cycle into an individual-based DGVM, LPJ-GUESS (v4.1, r10994), in order to improve the model performance with regard to observations of vegetation and soil N and P stocks and fluxes in comparison to the N-only (LPJ-GUESS-CN) model version. The new model version (LPJ-GUESS-CNP v1.0) includes soil organic P dynamics, P limitation of organic matter decomposition, P deposition, temperature and humidity-dependent P weathering, plant P demand and uptake, and P limitations to photosynthesis. Using the CNP version of LPJ-GUESS we also estimated global spatial patterns of nutrient limitation to plant growth, as well the temporal change in plant N and P limitation during the 20th and early 21st centuries, evaluating the causes for these temporal shifts.

We show that including the P-cycle significantly reduces simulated global vegetation and soil C and N stocks and fluxes, in particular in tropical regions. The CNP model simulation improved the fit to global biomass observations in relation to the CN simulation. The CNP model predicted predominant P limitation of plant growth in the tropics, and N limitation in the temperate, boreal, and high altitude tropical regions. The CNP model also correctly predicted the global magnitude (~ 50 PgP) and the spatial pattern of total organic P stocks. P limited regions cover less land surface area (46%) than N limited, but are responsible for 57% of global GPP and 68% of vegetation biomass, while N limited regions store a larger portion of total

carbons stocks (55.9%). Finally, the model showed that globally primary production limitation to N availability decreased and limitation to P increased from 1901 to 2018, with N being more responsive to temperature, and P to $CO_2$ changes. We conclude that including the P-cycle in models like LPJ-GUESS is crucial for understanding global-scale spatial and temporal patterns in nutrient limitation and improving the simulated carbon stocks and fluxes.

## 1 Introduction

Vegetation productivity is strongly limited by available nutrients, especially nitrogen (N) and phosphorus (P) (Van Der Heijden et al., 2008; Wieder et al., 2015; Zhu et al., 2016). Nutrient availability determines plant carbon storage and community composition (Quesada et al., 2012; Wieder et al., 2015). With rises in atmospheric $CO_2$ concentrations and its impacts on climate and plant physiology, such as $CO_2$ fertilization (Hickler et al., 2015; Walker et al., 2021), nutrient availability increases in importance as a factor limiting vegetation growth (Hickler et al., 2015; Johnson, 2006; Kou et al., 2020; Luo et al., 2004). Furthermore, anthropogenic impacts on the nutrient cycle may affect the community balance from one major nutrient limitation to another. For example, N deposition exacerbates P limitation (Peñuelas et al., 2013) while increases in fire frequency result in attenuation of P limitation, as N volatilizes and P is retained in the system (Butler et al., 2018).

A global analysis of terrestrial ecosystem N and P limitation showed that, excluding cropland, urban and glacial areas, only 18% of natural terrestrial land area was predominantly N limited, whereas 43% was predominantly P limited (Du et al., 2020). This pattern might occur across the entire tropical swath of the globe (Cunha et al., 2022; Vitousek, 1984; Wright, 2019), where highly weathered soils provide very little available P for plant growth. Global patterns of soil P availability indicate that it increases from the equator to the poles, as well as from lowlands to highlands in the tropical regions (Du et al., 2020). Soil characteristics also play a major role, with organic carbon and phosphorus content, parent material and sand content being the most important predictors of P availability (He et al., 2021b; da Silva et al., 2022). In these regions of relative low P availability, a tight coupling of P cycling between plants and the soil biota in these regions has been observed (Wilcke et al., 2019), as well as diverse plant P-acquisition strategies (Reichert et al., 2022). Fertilization experiments in these regions resulted in rapid changes to the plant communities, with increased productivity after P (Lugli, 2021), but also N additions, suggesting that both elements limit plant growth concurrently (Li et al., 2016; Wright, 2019). In light of this, understanding vegetation response to N and P availability and their global patterns is crucial to project the future carbon balance of the Earth (Fleischer et al., 2019; Leuschner et al., 2013).

For such large time (i.e. decades to centuries) and spatial (local to global) scale studies, the use of Dynamic Global Vegetation Models (DGVMs) has been an important tool (Prentice et al., 2007; Quillet et al., 2009; Shi et al., 2021). The inclusion of nutrient cycling and nutrient constraints of productivity in DGVMs has been carried out since the last decades, since models without nutrient cycles overestimate the potential future increase in ecosystem carbon storage under increasing $CO_2$ (Cramer et al., 2001; Fleischer et al., 2019; Haverd et al., 2020). Historically the N cycle has seen the most widespread

inclusion into DGVMs (Von Bloh et al., 2018; Fisher et al., 2010; Smith et al., 2014; Zaehle and Friend, 2010) due to the focus on the predominantly N-limited ecosystems from the higher latitudes of the northern hemisphere (Zhu et al., 2015). However, much tropical areas such as moist forests and Savannas, which account for a large portion of vegetation biomass and productivity, are predominantly P-limited (Du et al., 2020; Field et al., 1998). In addition, responses of the N and P cycles to environmental change can differ greatly due to different biogeochemical controls in each (Vitousek et al., 1997). In light of this, the P-Cycle was included recently in several vegetation models, such as ORCHIDEE (Goll et al., 2017), CASA (Wang et al., 2010), CLM (Yang et al., 2014), CABLE (Haverd et al., 2018), FUN (Allen et al., 2020), and CoupModel (He et al., 2021a), having as a common goal improved fit to gross primary production (GPP) and biomass observations, and improved model response to $CO_2$ fertilization. Indeed, compared to DGVMs which do not include the P-cycle, $CO_2$-induced biomass growth differed as much as 50% when simulating a low P site in the Central Amazon (Fleischer et al., 2019). In spite of this, these implementations rarely had a global extent nor evaluated the progression of N and P limitation during the last century. Also, a thorough investigation of which environmental factors can impact differently the N and P cycles is still absent.

Here we implemented the P-cycle into the DGVM LPJ-GUESS (Smith et al., 2001, 2014), including relevant processes into the vegetation and soil dynamics. We then executed global simulations of the model in CN (only the carbon (C) and the N cycles affecting the simulation) and CNP (C, N and P cycles affecting the simulation) versions (CN: N limited, CNP: N and P limited) in order to answer the following questions: (1) Does including the P-cycle improve model agreement to biomass and GPP observations? (2) What drives N and P limitation across climate zones? (3) How has that changed during the 20th and early 21st century? and (4) Which environmental factor change among CO2 concentration, N deposition, precipitation and temperature, are more relevant for N and P limitation change during this period?

## 2 Methods

### 2.1 General model description

The Lund-Potsdam-Jena General Ecosystem Simulator (LPJ-GUESS) DGVM (Smith et al., 2001, 2014) is an individual-based DGVM that simulates vegetation biome distribution and shifts, forest succession, disturbances by wildfires and biogeochemical cycles. The model includes, for example, individual representations of tree cohorts, a Farquar-based photosynthesis implementation, establishment and competition processes (light, water, nutrients, space), fire, and mortality which includes the effect of age and low growth efficiency. Each tree and grass individuals belong to a specific Plant Functional Type (PFT), which determines their properties by pre-defined trait parameters.

Previous versions of the model already include soil organic matter (SOM) dynamics following the CENTURY approach (Parton et al., 1993, 2010) with organic matter pools, and the N cycle (Smith et al., 2014). Organic matter enters the SOM model through vegetation litter input, which through tissue characteristics and traits, strongly influences soil N content and SOM decomposition rates. Simulated tree and grass individuals depend on mineral available N for growth. Their N demand

is the product of the leaf N concentration needed to optimize the carboxylation capacity of the canopy to maximize GPP and the size of other compartments of the individuals where their N concentrations follow the optimal leaf N concentration. Lower N uptake than demand reduces photosynthetic rates and drives increased relative allocation to roots, increasing N uptake capacity and strength in respect to other tree or grass individuals (Smith et al., 2014). The model has been widely evaluated globally (Wärlind et al., 2014) and regionally (Hickler et al., 2012).

## 2.2 P soil processes

The inclusion of the P-cycle soil processes in LPJ-GUESS followed closely the implementation of the CENTURY model structure for P (Parton et al., 2010), from which the N-cycle also was adopted. From the CENTURY P submodel we included the values for the C:P ratios of the slow, passive and active (microbial) organic matter pools and their variation according to the labile (inorganic) P pool.

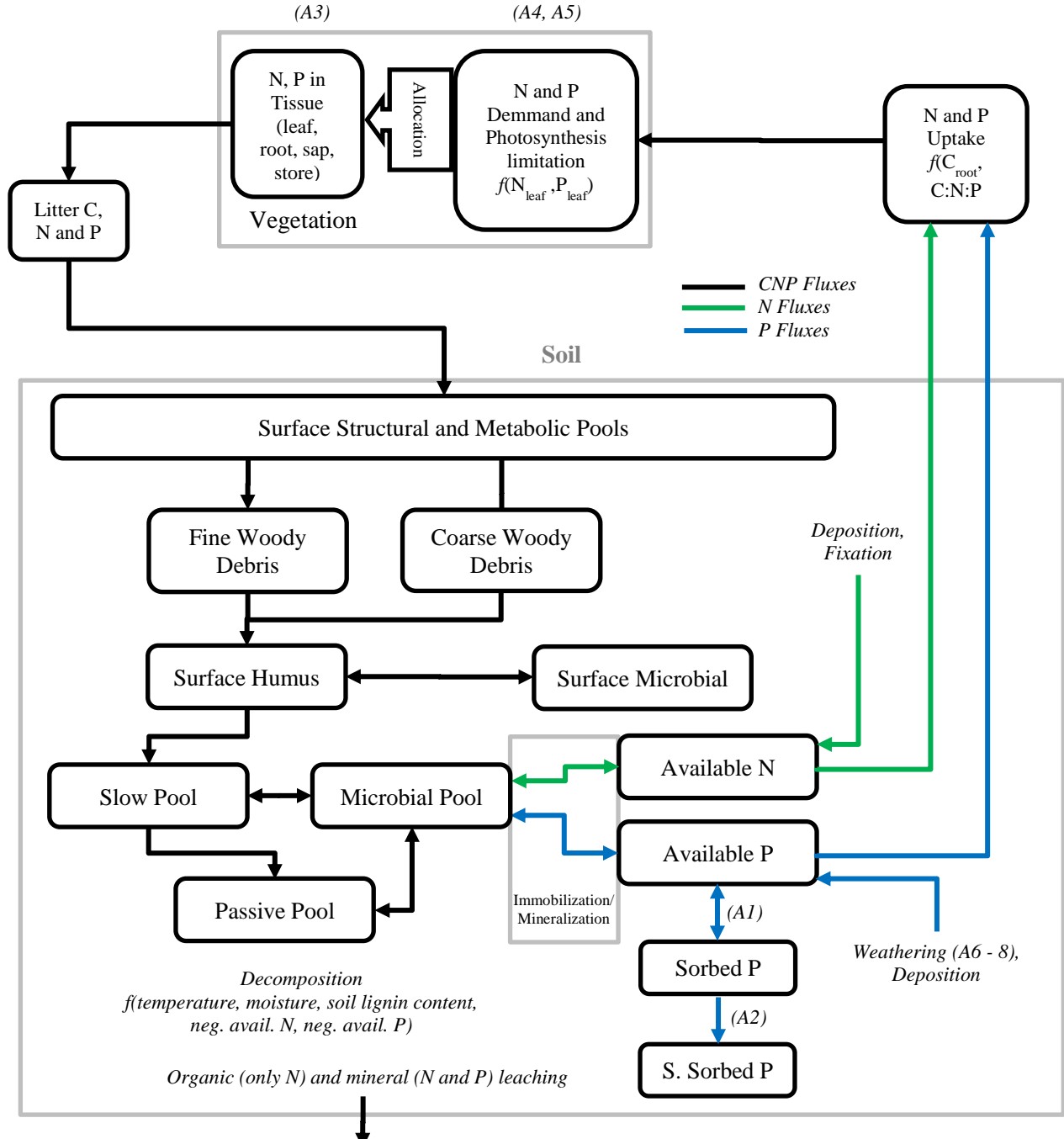

110

**Figure 1. Fluxes of C, N and P and modeled pools of soils and vegetation. Pools connected by a blue arrow are new for the phosphorus version of LPJ-GUESS. "A" in parenthesis refer to equations in the appendix for the related processes.**

An overview over soil and vegetation interaction scheme is given in Fig. 1. The P soil implementation was included into the

115  existing organic and inorganic matter pools established in the soil module developed for the N-cycle (Smith et al., 2014).

New organic matter pools exclusive to the P-Cycle were the available P pool (representing mineral P which plants can readily uptake), the sorbed P pool, and the strongly sorbed P pool (Fig. 1). For the sorption/adsorption processes, which determine the dynamics between the available, sorbed and strongly sorbed pools, we used the equations of the CASA-CNP model by Wang *et al.*, (2007, 2010). Here, the relationship between sorbed and available pools is based on the Langmuir equations (Bolster and Hornberger, 2007) and is calculated daily. The relationship between sorbed and strongly sorbed pools is proportional to the amount of sorbed P in the soil (Eq.A2). From Wang *et al.*, (2010) we also used the soil type specific parameters for equilibrium between sorbed and labile P ($K_{plab}$) and the maximum amount of sorbed P ($S_{pmax}$) (Table A1).

Similar as available N in the CN version of LPJ-GUESS, the available P pool can influence decomposition rates. If immobilization is larger than available pools, decomposition rates are reduced, meaning that available N or P may can limit decay rates. Inorganic leaching follows the same approach as the N cycle in LPJ-GUESS (Smith et al., 2014), but leached organic P is retained and considered mineralized, in order to reflect the tighter plant-soil coupling (Parton et al., 1988). Although in LPJ-GUESS mineral soil is represented up to 150 cm, divided by 15 layers, SOM is simulated as a bulk pool without explicit depth, with the temperature and moisture of the third mineral layer (30 cm) affecting SOM dynamics. The effect of fire on the P-Cycle is also implemented differentially than in N. While part of N contained in burnt plant tissue and litter (both fine and coarse woody debris) is volatilized and goes into the atmosphere, we consider burnt P from plant tissue and litter to be completely retained in the soil.

P weathering is a key process in P cycling, and was implemented following the empirical chemical weathering model (CWM) of (Hartmann and Moosdorf, 2011), with the soil shielding factor (Hartmann et al., 2014). The model (Eq.A6) is initialized with the global lithological map of the Earth at a 0.5x0.5 degree spatial resolution, containing the model's weathering parameters for each lithological class (Hartmann and Moosdorf, 2012). Daily weathering rates are calculated for each simulated patch using soil temperature and runoff derived from LPJ-GUESS. Since runoff is influenced by vegetation structure, our model thus provides a relationship between vegetation and weathering. A detailed description of the chemical weathering model can be found in the appendix.

**2.3 P in plant tissue and uptake**

Individual plants at each time step attempt to uptake P from the inorganic P pool (p_labile) in accordance to their P demand, which is calculated from each individual's optimal leaf C:P value. This plant functional type (PFT) specific value is estimated using the specific leaf area (SLA) parameter according to the equation S3 which was derived in turn using a global tradeoff relationship from the TRY database (Kattge et al., 2020). The amount of P each individual can take from the P_labile pool is limited by the total root area (determined using root biomass) and the Michaelis-Menten uptake dynamics, parameterized using the measurements from Silveira & Cardoso, (2004), resulting in a Vmax uptake of $1.48 \ 10^{-3} \ gPO_4 \ gC^{-1} \ d^{-1}$, and a half-saturation constant of $2.15 \ 10^{-7} \ kgPO_4 \ l^{-1}$ In addition, individuals also compete for the available p_labile in each simulated patch, following the same approach as for the previously implemented N competition in LPJ-GUESS. If the

individual is not able to uptake enough P to meet its daily demand, then it is considered P-limited (which does not exclude
being N-limited), leading in turn to a decrease in leaf to fine-root ratio (increase of relative root allocation) and a reduction of
photosynthetic capacity, both on a yearly scale. P is returned to the soil organic matter through plant death and/or senescence
of leaves and fine roots. In the case of the latter, a fixed 50% of P in leaves and fine roots is resorbed before transfer to the
fine litter organic pool.

## 2.4 P effects on photosynthesis

In each timestep, the carboxylation rate ($Vc_{max}$) of the individual is calculated as follows: 1. Without considering leaf
nutrient content ($Vc_{max}$), 2. Considering leaf N content ($Vc_{max,N}$), and 3. Considering leaf P content ($Vc_{max,P}$). In the CNP
version of GUESS, the smallest of 1, 2 and 3 is the value used for $Vc_{max}$, according to Liebig´s law of the minimum. More
detailed information on the calculations of $Vc_{max}$ and $Vc_{max,N}$ can be found in Smith *et al.*, (2014), equation C10. The
calculation of $Vc_{max,P}$ is based on Hidaka & Kitayama, (2013) for tropical trees. The equation (Eq.A4) uses the value of
active P, which is a fixed fraction of leaf P content (Eq.A5) and based on the metabolic P measurements from Hidaka &
Kitayama, (2013). By dividing $Vc_{max,N}$ or $Vc_{max,P}$ by $Vc_{max}$, we establish daily N and P limitation (0 - no limitation, 1 - full
limitation) on $V_{c,max}$ for each individual cohort:

$$Vc_{max,Nlim} = 1 - \frac{Vc_{max,N}}{Vc_{max,0}} \qquad \text{(Eq. 1)}$$

$$Vc_{max,Plim} = 1 - \frac{Vc_{max,P}}{Vc_{max,0}} \qquad \text{(Eq. 2)}$$

By normalizing with daily GPP (Eq. 3 and 4) over all cohorts ($N_{indiv}$) we get daily patch average $V_{c,max}$ limitations.

$$Vc_{max,Nlim\,(GPP)} = \frac{1}{\sum_{i=1}^{N_{indiv}} GPP_i} \sum_{i=1}^{N_{indiv}} (Vc_{max,Nlim,i} GPP_i) \qquad \text{(Eq. 3)}$$

$$Vc_{max,Plim\,(GPP)} = \frac{1}{\sum_{i=1}^{P_{indiv}} GPP_i} \sum_{i=1}^{P_{indiv}} (Vc_{max,Plim,i} GPP_i) \qquad \text{(Eq. 4)}$$

To get annual $Vc_{,max}$ limitations we normalize similar to Eq. 3 and 4 the daily patch nutrient limitations with total patch GPP
throughout the year. By normalizing through all the steps in calculating the annual nutrient limitations, we put a larger
influence on highly productive and probably larger individual cohorts. When the nutrient limitations are known it is also
possible to determine the magnitude of co-limitation by calculating

$$Vc_{max,NPlim} = \frac{Vc_{max,Nlim(GPP)} - Vc_{max,Plim(GPP)}}{Vc_{max,Nlim(GPP)} + Vc_{max,Plim(GPP)}} \qquad \text{(Eq. 5)}$$

This results in positive values for predominant N limitation, and negative values for predominant P limitation. In order to account for productivity differences between gridcells, global spatial averages of $Vc_{max,NPlim}$ are weighted by gridcell values of GPP.

## 2.5 Driving data

Our simulations were run for the years 1901 to 2018. For temperature, precipitation, and radiation we used daily sums or averages of the CRU JRA 2.0 dataset Harris et al (2021); For $CO_2$ concentrations we used yearly values from (IPCC, 2003). N deposition was based on the ACCMIP dataset of multi-year averaged N historical values (Fig. A2a) on a 0.5x0.5 degree spatial resolution, ranging spatially from 0.13 to 105.62 kg ha$^{-1}$ y$^{-1}$ since 2005 (Lamarque et al., 2013), and is the sum of total
inorganic bulk and dry deposition of $NH_4$ and $NO_3$. P deposition is taken from a gridded dataset with 2x2 degree spatial resolution (Fig. A2c) with fixed monthly averages, ranging spatially from 0 to 65.23 kg ha$^{-1}$ y$^{-1}$ (Chien et al., 2014), and is the sum of total inorganic bulk and dry $PO_4$ deposition, and including total soluble particles. We consider only N deposition to have a historic trend (Fig. A1), with P deposition unchanging along our historic time period (1901 – 2018). This is assumed due to the time scale here evaluated (100 years) since from an aerosol deposition perspective, human perturbations
to the N cycle far exceed that of any other biogeochemical cycle (Mahowald et al., 2017). N aerosols are estimated to have increased by approximately 250% over pre-industrial conditions and over much of the industrialized regions of the Northern Hemisphere (Kanakidou et al., 2016). Main emission sources of N are from burning of fossil fuels, and the use of fertilizers in agriculture.

## 2.6 Model protocol, scenarios and evaluation

In order to evaluate the implications of adding the P-cycle into LPJ-GUESS, we execute the model in the CN and the CNP versions and calculate global averages of C, N, P stocks and fluxes, and we provide global maps of potential natural vegetation, on a 0.5x0.5 degree grid covering the ice-free surfaces, averaging a part of the simulation time period (2005 - 2015). This time period was chosen in order to improve validation possibilities. In each gridcell, 15 patches of vegetation
were simulated, in which plant individuals compete for resources. In order to represent larger scale disturbances, individual patches are destroyed randomly with an interval of 100 years. The fire module we used for our simulations was the GLOBFIRM (Thonicke et al., 2001).

We then compare the simulated results to values from the literature. In order to validate the spatial patterns of the CN and CNP versions, we compare simulated vegetation biomass and GPP to the global biomass maps of ESA CCI (Santoro et al.,
2021) and GOSIF GPP (Li and Xiao, 2019) respectively. The Solar-induced fluorescence based (GOSIF) dataset is preferred

instead of other methods (e.g. MODIS GPP - Light-use efficiency or FluxCom - machine learning) due to its lower susceptibility of saturation in lower latitude regions (Pickering et al., 2022). We also compare our global soil P stocks to the empirically derived map from (He et al., 2021b) and produce global maps of $Vc_{max}$ N and P limitation, and which is dominant for each gridcell.

In order to analyze the global changes of N and P limitation of vegetation productivity from 1901 to 2018, we plotted the global average N and P limitation for each year, and calculated the trends of average limitation between the periods 1901-2018 for each grid cell. Finally, aiming to evaluate the climatic and edaphic drivers of global N and P limitation changes, we conducted a factorial experiment in which temperature, and precipitation were fixed at mean and detrended 1901-1931 levels, and replicated until 2018. $CO_2$ concentration was fixed at 1901 measurements (295.3 ppm) and N deposition fixed at

2 kg ha$^{-1}$ y$^{-1}$ N. The scenarios considered for these factorial runs were in the following denoted as "Allfixed" (all four drivers fixed), "CO2var" (only $CO_2$ concentrations varying according to observations), "Tempvar" (only temperature varying according to observations), "Precvar" (only precipitation varying according to observations) and "Ndepvar" (only N deposition varying according to observations).


## 3 Results

### 3.1 Global CNP stocks, fluxes and evaluation

In general terms, including the P-cycle and vegetation limitations to P availability reduced plant productivity and biomass, as well as the other element stocks and fluxes (Table 1). In relation to the CN version, CNP reduced GPP by 7% and biomass

by 19%. With regards to observations, both model versions CN and CNP values fell within ranges for GPP, NPP, Veg C, Veg N, Litter+Soil C and overestimated Litter+Soil N. Values for Litter+Soil N, N leaching, Litter+Soil P and P labile were underestimated, but in general corresponded to the order of magnitude (Table 1).

| Variable | Units | CN | CNP | literature-based | refs. |
|---|---|---|---|---|---|
| GPP | Pg C yr-1 | 133.2 | 123.7 | 108. . .159 | 1,5,10,17, 20 |
| NPP | Pg C yr-1 | 61.4 | 56.7 | 56. . .63 | 5,10,13,17 |
| Veg C | Pg C | 550.3 | 444.7 | 470. . .650 | 8,13, 9 |
| Litter+Soil C | Pg C | 1502.6 | 1474.1 | 1270. . .2010 | 8,12 |
| Veg N | Pg N | 3.3 | 2.8 | 3.5 | 14 |
| Litter+Soil N | Pg N | 126.8 | 127.2 | 95. . .118 | 12,14 |
| N uptake | Tg N yr-1 | 683.9 | 644.7 | | |

| | | | | | |
|---|---|---|---|---|---|
| Net N min | Tg N yr-1 | 657.1 | 648.1 | | |
| N leaching | Tg N yr-1 | 33.7 | 35.9 | 52 | 2 |
| N fixation | Tg N yr-1 | 56.9 | 51.2 | 100…290 | 3,6 |
| Veg P | Pg P | - | 0.2 | | |
| Litter+Soil P | Pg P | - | 51.9 | 40.6 … 89 | 18, 22 |
| P uptake | Tg P yr-1 | - | 53.6 | | |
| Net P min | Tg P yr-1 | - | 29.8 | | |
| P leaching | Tg P yr-1 | - | 1.4 | | |
| P weathering | Tg P yr-1 | - | 1.088 | 1.144 | 21 |
| P labile (PO4) | Pg P | - | 2.11 | 3.6 | 22 |

**Table 1. Mean Global C, N and P stocks and fluxes for the N-only (CN) and N with P (CNP) simulation runs. Soil C, N and P refers to total organic matter pools minus litter pools; Litter refers to pools of shed leaves, fine roots, fine and coarse woody debris and is added through vegetation tissue turnover. References: 1. Beer et al. (2011); 2. Boyer et al. (2006); 3. Cleveland et al. (1999); 4. Cleveland and Liptzin (2007); 5. Demarty et al. (2007); 6. Galloway et al. (2004); 7. House et al. (2003); 8. IPCC (2007); 9. Ito et al. (2004); 10. Ito et al. (2011); 11. Piao et al. (2010); 12. Post et al. (1985); 13. Saugier and Roy (2001); 14. Schlesinger et al. (1997);**
**15. Schultz et al. (2008); 16. van der Werf et al. (2010); 17. Zhang et al., 2009; 18. He et al. 2021; 19. Santoro et al. 2019; 20. Li and Xiao 2019; 21. Hartmann et al. 2014. 22. Yang et al. 2013. The time period for the simulated averages is 2005 - 2015.**

Global patterns of biomass and productivity changed noticeably between the CN and CNP model versions (Fig. 2).
Activating the P-cycle resulted in significantly lower biomass, particularly in the tropical region, but also less prominently in Northeastern USA and Northern Europe (Fig. 2c). GPP also decreased most strongly in the tropics with the inclusion of the P-cycle, but also showed changes in temperate and boreal zones, with even increases in southern Finland and western Russia (Fig. 2d).

**a**                    Biomass                              **b**                    GPP

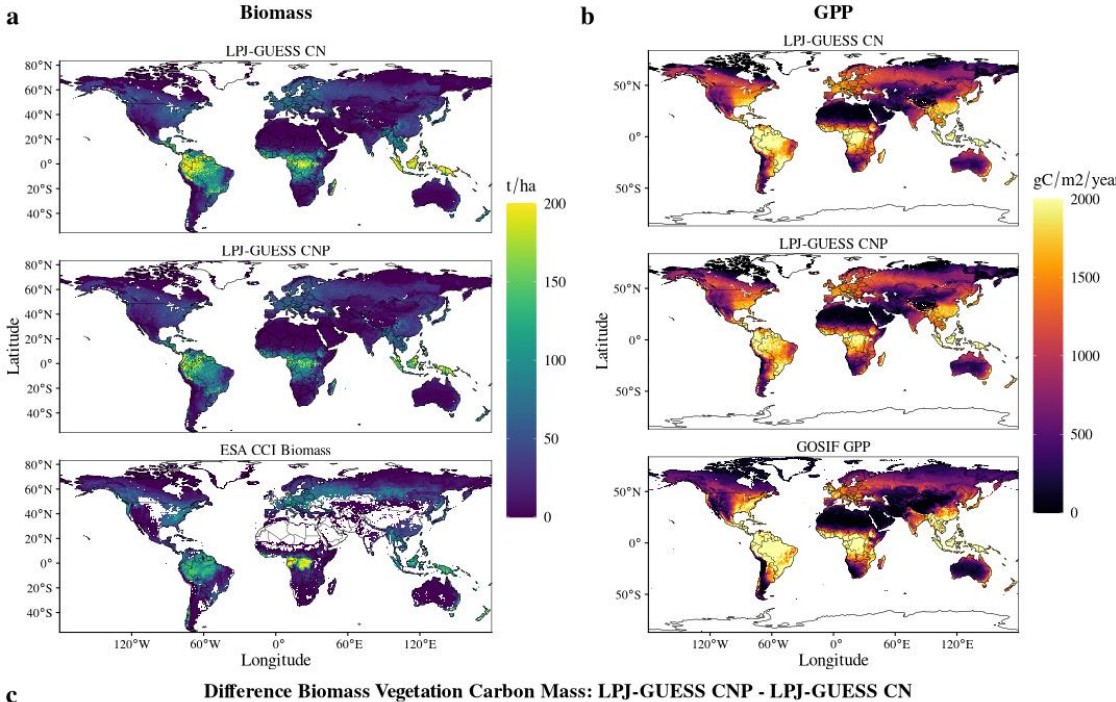

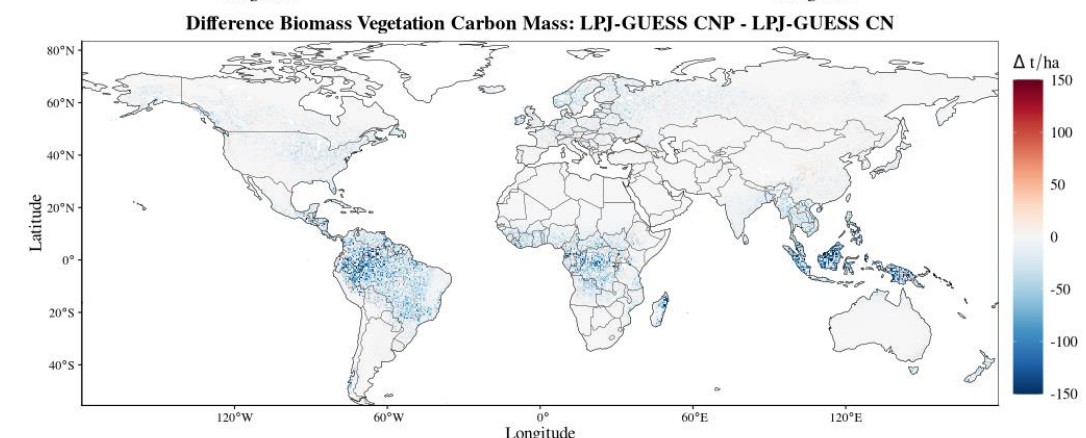

**c**    Difference Biomass Vegetation Carbon Mass: LPJ-GUESS CNP - LPJ-GUESS CN

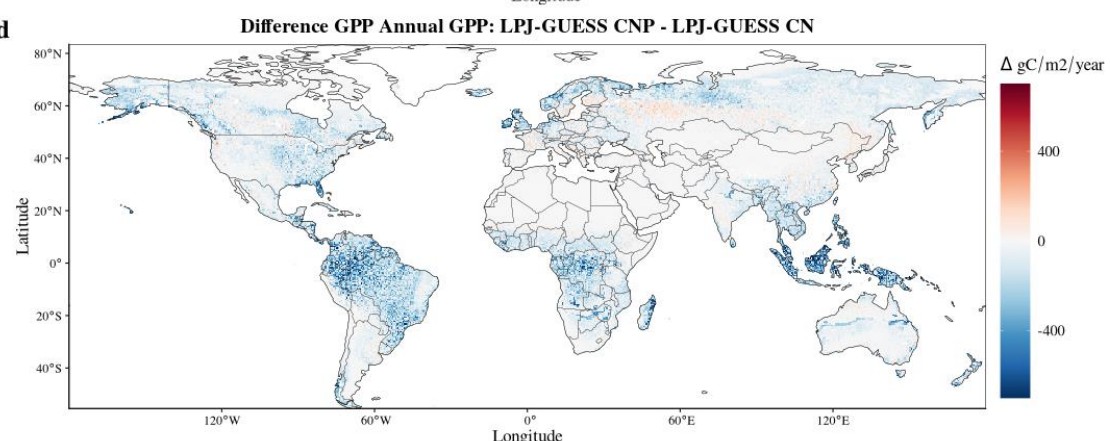

**d**    Difference GPP Annual GPP: LPJ-GUESS CNP - LPJ-GUESS CN


**Figure 2: Global maps of biomass (a) and GPP (b) for the CN, CNP model simulations and evaluation datasets (Biomass: Santoro *et al.*, (2021); GPP: Li & Xiao, (2019)). (c) and (d) maps indicate the differences between the two model setups. Negative values indicate lower estimates by the CNP version.**


Estimates of model error and fit to the reference ESA CCI biomass map showed that including the P-cycle improved simulated results of global vegetation C stocks. In relation to the reference data, we observed a general reduction of model error from the CN to the CNP version (Table 2). This resulted in less deviation from the reference ESA CCI biomass map (Fig. 3a). With regards to GPP, the already evident underestimation of GPP in relation to the GOSIF data was exacerbated in

the CNP version, as expected due to the inclusion of an extra factor of limitation to productivity. This resulted in a worsening of fit to observations from CN to CNP, when considering the GOSIF dataset as reference (Fig. 3b).

| Metric | Biomass (ESA CCI) vs: | | GPP (GOSIF) vs: | |
|---|---|---|---|---|
| | CN | CNP | CN | CNP |
| RMSE (Root Mean Squared Error) | 43.35 | 32.91 | 464.22 | 527.96 |
| r2 (Coefficient of Determination) | 0.50 | 0.52 | 0.80 | 0.77 |

**Table 2. Statistical measures of the comparisons of model runs to the ESA CCI Biomass and GOSIF GPP maps. Calculated using the R DGVMTools package (github.com/MagicForrest/DGVMTools).**

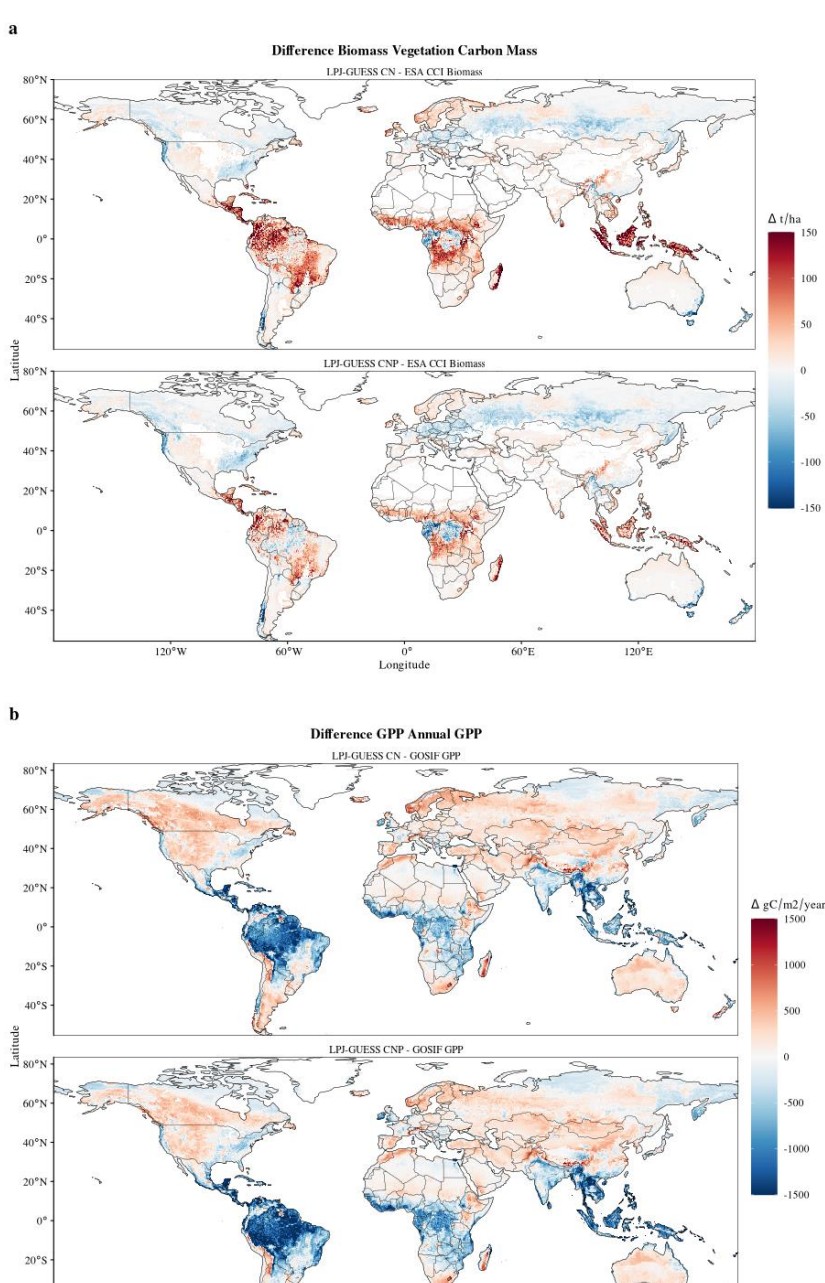

**Figure 3: Differences of model simulations to global biomass (a) and GPP (b) reference maps. Red are model overestimations, blue are underestimations.**

### 3.2 Global simulated spatial patterns of N and P limitation and P stocks

Spatial averages of N limitation for the period 2005 - 2018 showed lowest values for the tropics, and P limitation showed a less predictable pattern (Figs. 4a and b). The global plots of predominant N and P productivity limitation resulted in the expected global spatial patterns from other studies. For example, similarly to the map produced by (Du et al., 2020), our simulated map showed predominant N limitation in higher latitudes, dry areas and higher elevations and predominant P limitation throughout the tropical region (Fig. 4c). Differently to Du *et al.*, (2020), our map resulted in less gridcells being predominantly P limited, 27%, with the rest and majority being N limited.

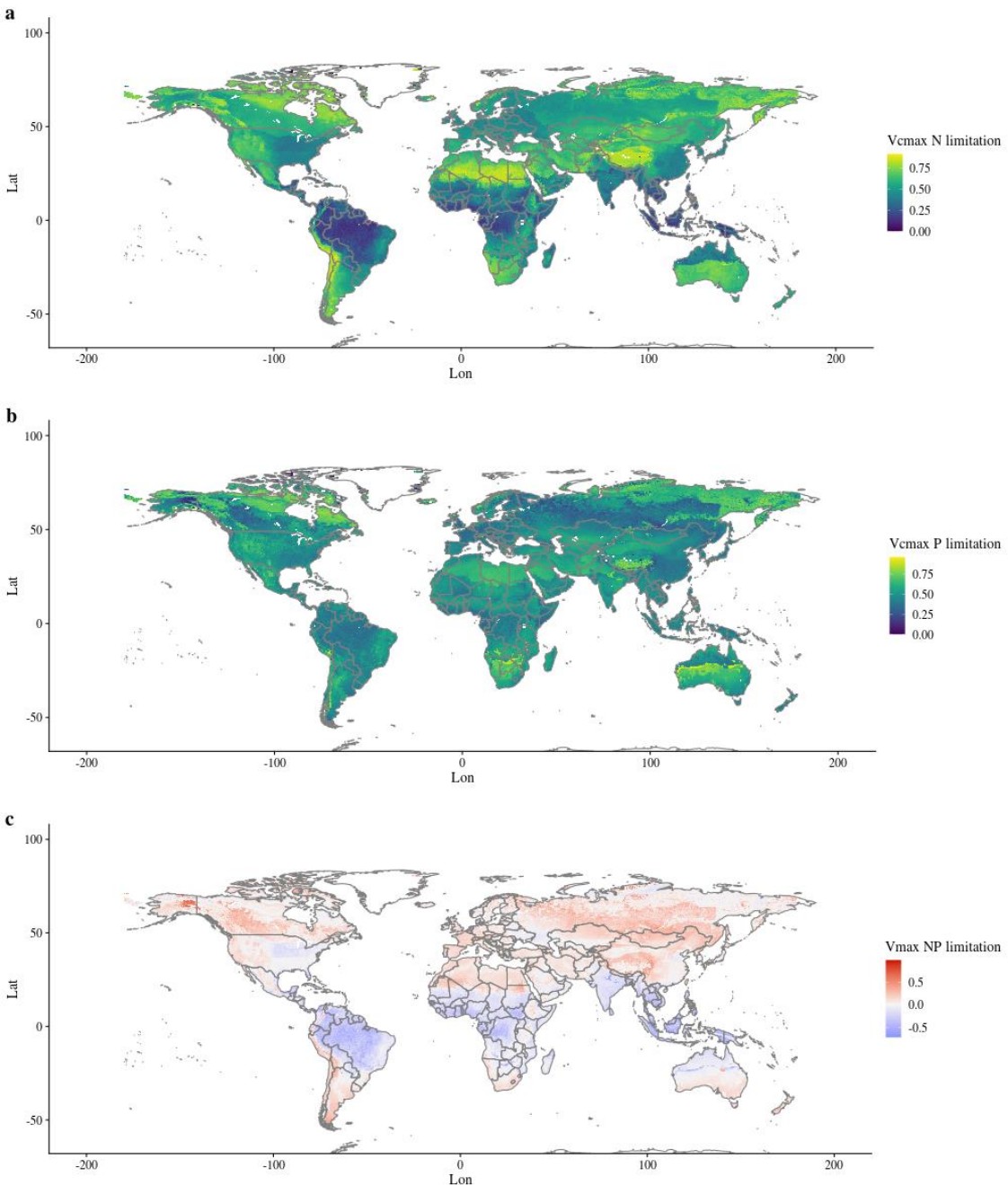

**Figure 4: Global maps of (a) N and (b) P limitation, 0 represents no limitation to Vc$_{max}$ (unitless), and 1 complete limitation. (c) Map of predominant N or P limitation calculated using Equation 3, with positive values being predominantly N limited, and negative values predominantly P limited. Values for each gridcell are averages from 2005 - 2018.**


Simulated organic stocks of P (Litter+Soil P) showed a very similar spatial pattern as the field based estimates from He *et al.*, (2021), with the highest soil organic P stocks in Canada and Russia (Fig. 5a). The model also correctly predicted higher P stocks in tropical high elevations such as the Andes, but failed to predict the high stocks of the Tibetan Plateau. Soil labile

P stocks (Fig. 5b) look similar to a combination of P deposition and weathering (Fig. A2d), and is influenced by water runoff. N fixation results showed an expected spatial pattern of high values in the tropical regions and lower elsewhere (Fig. A2b).

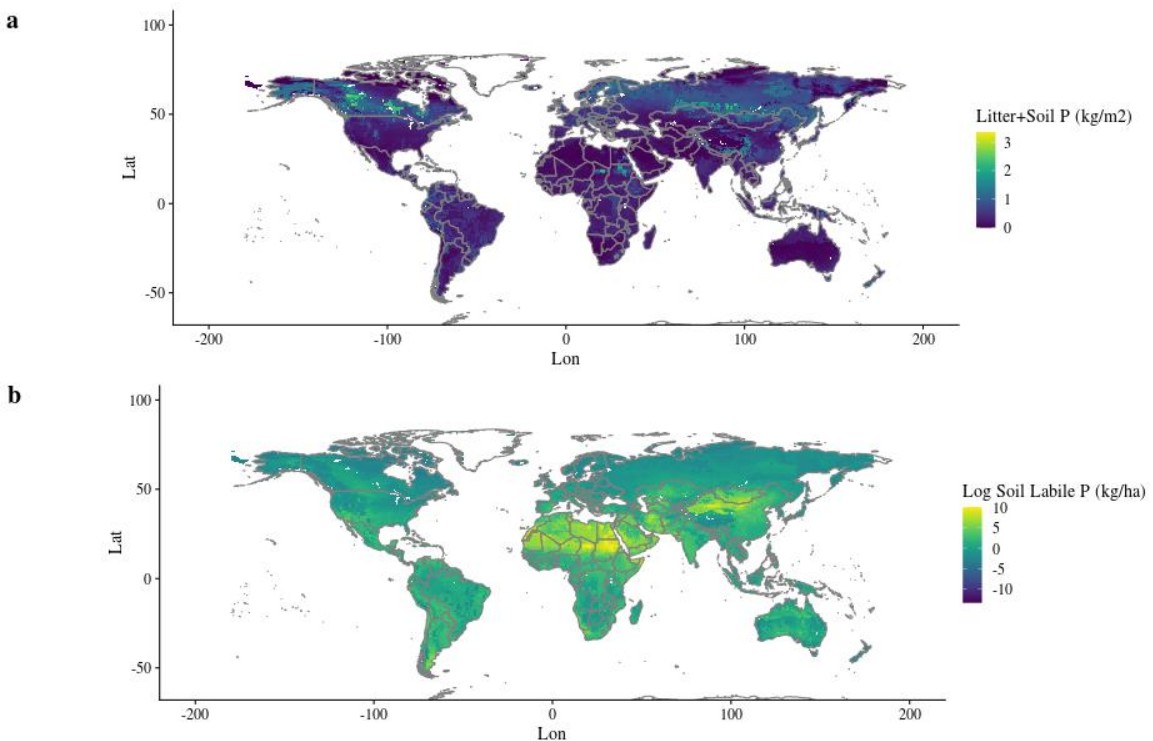

**Figure 5. Global maps of soil organic (a) and labile (b) P stocks. Soil P refers to total organic matter pools minus litter pools; Litter refers to pools of shed leaves, fine roots, fine and coarse woody debris and is added through vegetation tissue turnover.**

### 3.3 Global historic trends of N and P limitation

Average values for N and P limitation were shown to have shifted in our simulation, in response to the environmental changes, which include temperature, precipitation, $CO_2$, and N deposition, from 1901 to 2018. The trends of nutrient limitation during this time period indicate that 71% of the grid cells showed a reduction in N limitation, while 64% showed

an increase in P limitation (Fig. 6a). This resulted in a global trend from 1901 to 2018 where N limitation decreases and P limitation increases, overtaking N limitation as the most dominant element, when accounting for gridcell GPP (Fig. 6b).


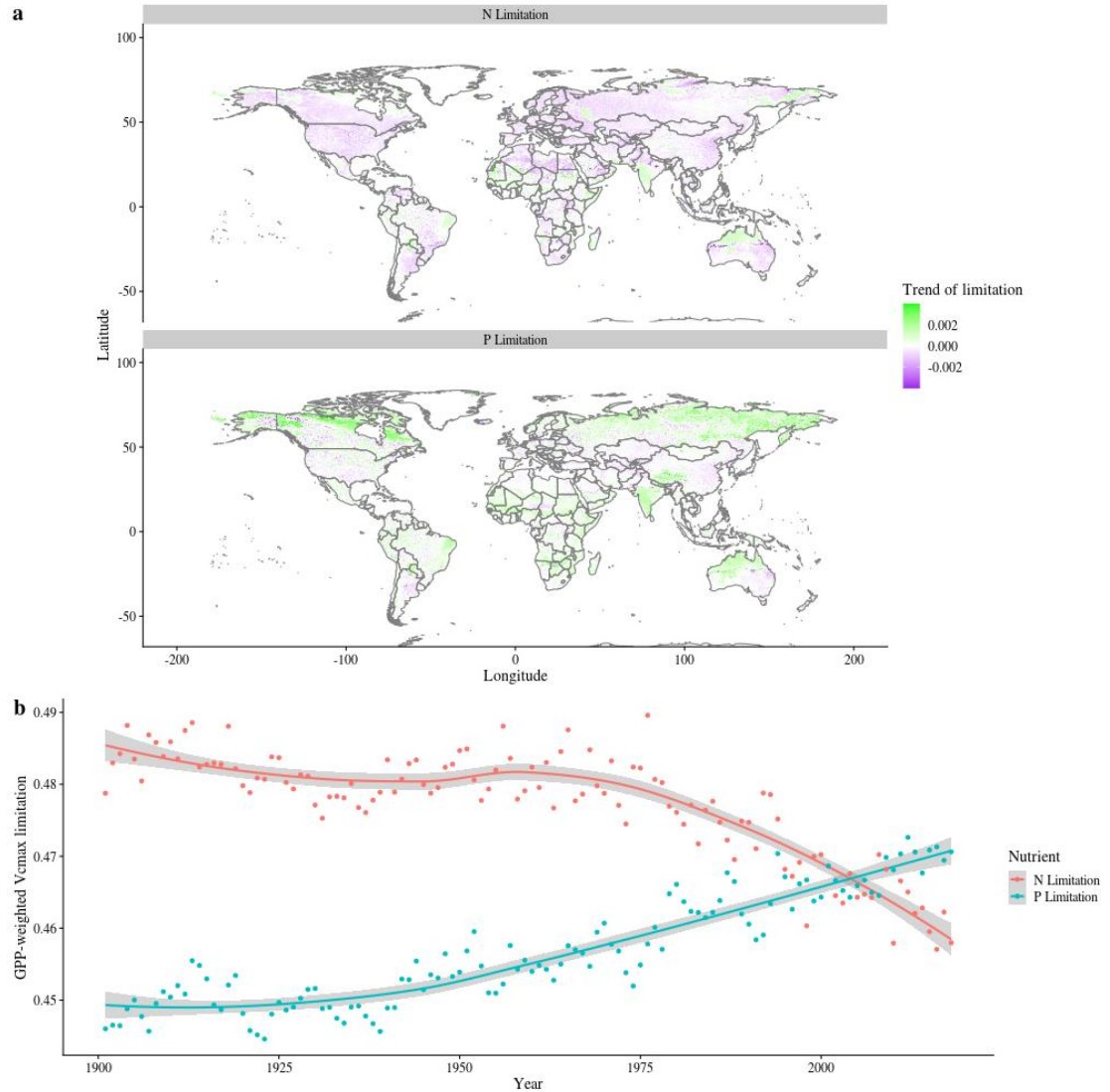

**Figure 6. Global simulated trends of N and P Vc$_{max}$ limitation from 1901 to 2018. (a) N and P limitation trends, per gridcell for the 1901-2018 period. Negative (purple) values indicate reduction of nutrient limitation, and positive (green) values indicate increases of nutrient limitation. (b) Time series of GPP-weighted N and P Vc$_{max}$ limitation, each point represents the average global**
**limitation for N or P for all gridcells.**

When isolating the individual climatic and edaphic drivers for the model in the factorial analysis, we observed that precipitation and N deposition changes from 1901 to 2018 had little impact on global modeled Vc$_{max}$ limitation, even though

N deposition increases strongly from 1901 to 2018 (Fig. A1). The inclusion of temperature variation reduced significantly N

limitation, but had only a weak effect on P limitation. Changes in $CO_2$ concentrations increased both limitations, particularly P (Fig. 7).

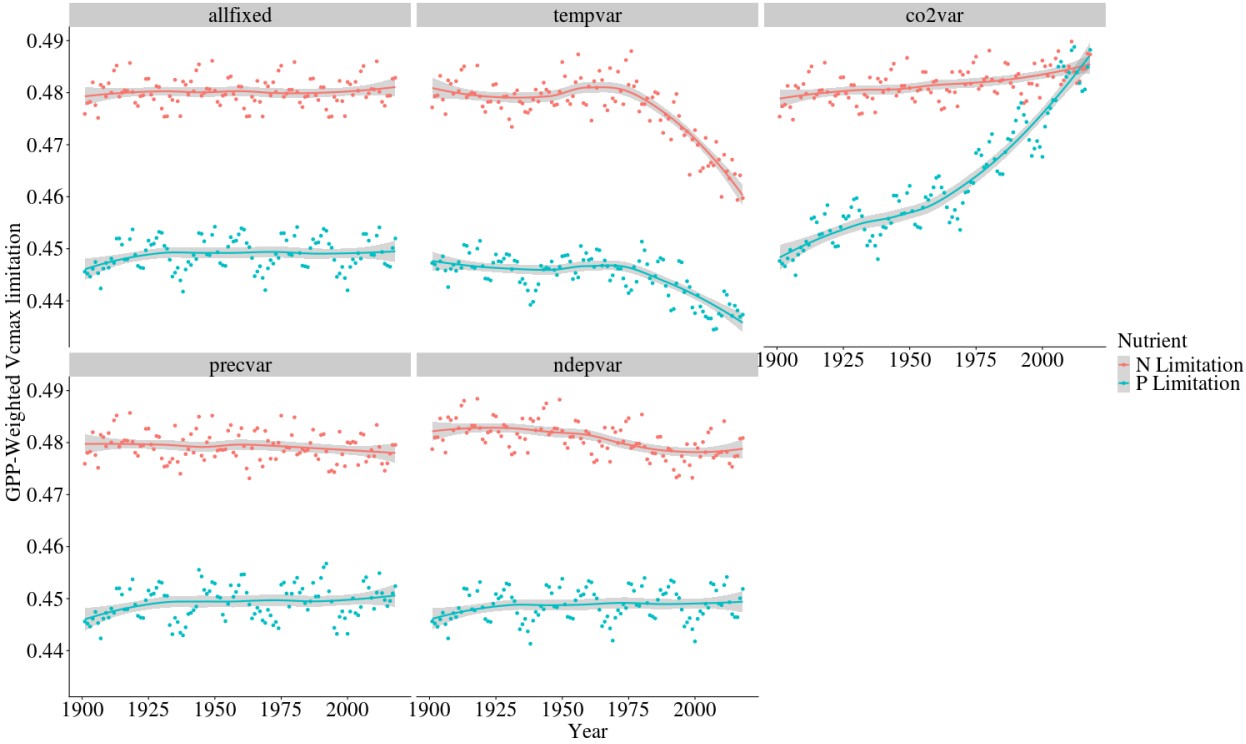

**Figure 7. Global simulated trends of N and P Vc_{max} limitation resulting from the factorial experiment.**


## 4 Discussion

With our implementation of the P-cycle and P limitations to photosynthesis into LPJ-GUESS we were able to (1) reduce biomass overestimations in comparison to the only-N model; (2) reproduce the expected global pattern of N and P limitation to photosynthesis, both the latitudinal as well as the tropical altitudinal gradient; (3) reproduce the global geographical

pattern of soil organic P stocks, and fluxes of P within the magnitude of global total values; and (4) provide a glimpse into the global trends of N and P limitation to vegetation productivity for the last century, and which environmental factors are driving these trends.

## 4.1 Model evaluation and global patterns

The inclusion of the P-cycle in LPJ-GUESS improved slightly model fit compared to the CN version for biomass (Table 1, Fig.s 3c, 4a). In the case of GPP, reasons for the underestimation in relation to observations for the CN and CNP versions lie partially with the model's representations of light use efficiency and light interception (i.e. FAPAR calculation) as well as the choice of radiation input dataset. These may play a role here and contribute considerable to uncertainties. In addition, divergences between major global reference datasets of GPP (e.g. FLUXCOM and MODIS) hinder validations of process-
based models such as the one used in this study (Zhang and Ye, 2022).

Adding a limiting factor to photosynthesis, was expected to drive both C processes down, as was the case of our CNP simulation. The scale to which this reduction occurred, 7% for GPP and 19% for biomass globally, was not very high, given that according to Du *et al.*, (2020) 43% of the natural terrestrial area is predominantly limited by P. Although in that study, N or P limitation was defined as nutrient resorption efficiency while in ours it is $Vc_{max}$ limitation, our measured percentage of P
limited regions is remarkably close, at 46% (Table A2). Further, these predominantly P-limited gridcells are disproportionately important for the vegetation carbon cycle. 57.7% of global GPP is located in these P-limited regions, as well as 62.6% of NPP and 68.4% of vegetation biomass. With regards to total carbon stocks however, N-limited regions are more relevant than the tropics, storing 55.9%. This is due to the large soil organic C storage of N limited regions, 65.8% of the global total.


## 4.2 Soil Organic Matter Dynamics and the P-Cycle

The presence of a detailed soil organic matter (SOM) dynamics module in LPJ-GUESS allows an in-depth exploration of the links between vegetation and soil processes, and the implementation of LPJ-GUESS-CNP reveals the interactions between the N and P cycles. For instance, the influence of mineral P in soil decomposition, empirically representing bacterial nutrient
demand, also reduced net N mineralization (Table 1). P-limitation can thus exert an important role even in predominantly N-limited areas, since P can limit decomposition in areas where N limits photosynthesis.

Whether our simulated global organic and labile P stocks are realistic is still uncertain, due to large discrepancies between estimates of different authors, as well as varying methods. He *et al.*, (2021) calculated 26.8 ± 3.1 (mean + standard deviation) Pg of total P for soils up to 30 cm depth, and 62.2 ± 8.9 Pg for 30-100 cm, while Yang *et al.*, (2013) estimated
40.6 ± 18 Pg P for 0-50 cm. Our simulated total P stocks are located within this range, at 53 Pg P. However, since LPJ-GUESS-CNP does not consider soil depth for SOM, comparisons between simulated and empirical estimations here are difficult. Future versions of LPJ-GUESS are in development which include a definition of an explicit depth of SOM. Regarding simulated labile P stocks, our 2.45 Pg P are within the global estimations of 3.6 ± 3 Pg P (0 - 50 cm) according to Yang *et al.*, (2013), based on the Hedley P availability. Other methods such as Olsen P for global P plant available stocks,
are with 0.319 ± 0.022 Pg P (0 - 20 cm) significantly lower, but may be underestimating phosphorus stocks available for

plants. New estimations using the same method as Yang *et al.*, (2013) but with a much larger dataset resulted in even higher labile P estimations (He et al., 2023). The estimations by Yang *et al.*, (2013) correspond to a sum of labile inorganic plus labile organic P, which results in a pool of P which plants have access using direct and biomineralization pathways. These simplifications are suggested as adequate for models such as ours with a daily time step (Yang et al., 2013). We thus justify the absence of biomineralization processes in our model through our larger plant available P pool.

The inclusion of a P weathering process which accounts for precipitation and temperature in our model is an advancement due to several aspects. First, the chemical weathering model (CWM) allows not only to account for P release, but also other important plant nutrients such as Potassium (K), Calcium (Ca) and Iron (Fe) for future implementations. Second, the CWM provides a direct link between lithologies and plants, providing a framework to test hypotheses regarding the link between geodiversity and plant structural diversity. Third, in LPJ-GUESS, forest structure has an impact on hydrology, since runoff is influenced by the amount of intercepted precipitation and evapotranspiration, and this in turn is affected by total patch leaf area index. Since vegetation affects runoff, it has a significant impact on chemical release, because runoff is one of the variables in the CWM. We suspect that the effect of vegetation structure on runoff is the main reason why total P release from the model differs from observed P-release (Table 1), which is based on the same CWM, however using fixed temperature and runoff datasets. Thus, a DGVM which includes a CWM is the ideal tool to test hypotheses regarding the effects of climate change on future chemical weathering. We suggest that future DGVM developments in which chemical weathering plays an important role take advantage of this CWM to estimate the influence of vegetation in weathering. In addition, the CWM´s parameters were determined by a limited set of observations of rock P content (Hartmann et al., 2014; Hartmann and Moosdorf, 2011), potentially hampering global estimations. Therefore additional sampling campaigns, particularly in P limited regions, may significantly improve the CWM´s P release estimations and consequently our model results.

### 4.3 Spatial patterns of global NP limitation

While in most areas one nutrient limitation may clearly dominate over the other, in many regions co-limitation may also be very high (Wright et al., 2018). For instance, in many dry regions such as in Western China or in Australia both nutrients are severely limiting and the dominance of one factor over another does not mean that the non-dominant factor is not limiting. The high N fixation rates in the tropics (Fig. 5b) in our model are most likely the main driving factor for a predominant P-limitation there. The Indian subcontinent however, has low rates of fixation but a predominant P-limitation occurs due to high N deposition (Fig. 5a). Europe and North America however are still predominantly N-limited despite also exhibiting high N deposition, although in peak deposition areas of these two continents, the American Midwest and the Netherlands/Northern Germany/East Britain (Fig. A3), the predominant limiting element is phosphorus. Such shifts of commonly N-limited regions to P-limitation have indeed been reported, such as southern Sweden (He et al., 2021a). Regarding stocks and fluxes of modeled P, the choices of parameters may play a significant role, particularly those related to

soil decomposition, sorption/desorption, weathering, deposition, plant P demand and uptake, among others. Parametrization

and validation data for the P-cycle is scarcer than for the N-cycle, as can be seen for example from global plant trait databases (Kattge et al., 2020). Thus, increased effort in collecting field data on plant and soil P would be invaluable to test and improve current P-enabled DGVMs.

The maps produced by LPJ-GUESS in Figs. 4a and b indicate a general pattern of global co-limitation of N and P, of varying degrees. Simulated patterns of N and P limitation follow broadly that of CASA-CNP from Wang *et al.*, (2010), which is also

a process-based model and produces maps based on limitation to NPP as our study. Their study however shows predominant N limitation patterns for India and Australia, which in our approach had P as the main limiting factor. This and other differences may arise from a weathering input in CASA-CNP which did not consider temperature or runoff. Climatic factors are crucial in weathering, since even P rich substrates may not render significant amounts of nutrients if the climate is too dry or cold, or the lithology is shielded (Hartmann et al., 2014). Another approach, by Du *et al.*, (2020) also shows broad

similarities with our patterns, but are not directly comparable due to a distinct definition of N a P limitation, based on resorption efficiencies and the use of an empirical predictive model. Their NP limitation maps show greater P limitation in North America, Europe and Central Asia, but agree with our maps regarding the very high N limitation areas of Eastern Russia and the Tibetan Plateau, and unexpected predominant P limitation in many temperate areas, suggesting that P limitation is more widespread than previously thought (Hou et al., 2020).


## 4.4 Temporal trends of global NP limitation

The simulated global trends of N and P vegetation productivity limitation follow other global or regional studies that indicate an increasing role of P as a limiting factor (Li et al., 2016) and decreasing N limitation. Our simulated trends of N and P limitation should be considered here conservatively, since no land-use and fertilization aspects were considered.

Nevertheless, using our factorial simulation runs were able to identify which environmental factors during the period of 1901-2018 might have played a role in the trend of N and P limitation.

The drivers which affected N and P limitation to productivity were not the same, which is expected since the factors affecting the individual processes of the N and P cycles are different. Globally and on average, we have identified that N limitation decrease is predominantly driven by temperature, while P limitation increase is dominated by $CO_2$. In addition to

the temporal average global values of Fig. 7, we also present the spatial pattern of the trend slope for each factorial scenario, which indicates where nutrient limitation was affected by the varying factors (Fig. A4). The spatial pattern for the simulation experiment with temperature variation only (tempvar) scenario, in which N limitation is predominantly affected, shows that N limitation is increasing across most of the affected gridcells whereas for P there is a mixed signal with decreasing and increasing P limitation, resulting in almost no effect. A probable cause for a lack of strong average global effect of

temperature on decreasing P limitation was increases in P lim. for the boreal regions (Fig. A4). In spite of this strong increase in P limitation for the boreal regions, those areas however remained predominantly N-limited at the end of our

simulations. Decreases of C, N and P total soil stocks and increases of mineralization rates with increasing temperatures (Fig. A5) may have had a strong role in N limitation decrease, but less for P (Fig. 7).

The strong impact of $CO_2$ on both $Vc_{max}$ N and P limitation represents the increasing role of nutrients in limiting $CO_2$ fertilization. The stronger impact of $CO_2$ on P than on N limitation can be clearly inferred through the soil organic matter dynamics, as organic stocks of C and N increase and P decrease (Figs. A6a-c), and only N has increasing mineralization rates in time (Fig. A6d). N limitation increases due to $CO_2$ on the other hand were offset in our simulations by strong rises in nitrogen fixation rates (Fig. A6g). In LPJ-GUESS nitrogen fixation is modeled following Cleveland *et al.*, (1999), based on evapotranspiration. Since in our model evapotranspiration strongly depends on crown area and leaf biomass, which increases due to $CO_2$, the model provides an indirect path from which plants can invest carbon to alleviate N deficiency. Indeed, increased symbiotic nitrogen fixation, indicated by larger nodules, number, nitrogenase activity and plant N content has been commonly reported in plant cultivars in response to elevated $CO_2$ experiments (Leakey et al., 2009; Rogers et al., 2009; Xu et al., 2017). In addition, the proportion of fixed N in relation to soil-derived of resorption-derived has been seen to increase from ambient to elevated $CO_2$ (Li et al., 2017). It is reasoned thus that enhanced photosynthesis provides extra C sources for improving nodule function and $N_2$ fixation. For P there is no such pathway in our model for increased carbon availability to improve nutrition. However, plants are known to use several strategies investing carbon to acquire P (Lugli et al., 2020; Reichert et al., 2022; Smith and Smith, 2011; Stock et al., 2021). For instance, mycorrhizal associations - elevated $CO_2$ experiments indicated that ectomycorrhizal fungi (EMF)-mediated plants are able to gain more biomass in relation to arbuscular mycorrhizal (AMF) fungi, since the former can mine nutrients from organic matter (Terrer et al., 2021). Consequently, boreal regions in which EMF are abundant may be less limited than our studies suggest, although N (which is limiting in these environments) is necessary for producing the enzymes required for P acquisition. Also, plants are able to invest C into P acquisition using phosphatase and organic acids, and this investment may also increase under $eCO_2$ (Margalef et al., 2017). Therefore, we may be overestimating the increase of P limitation due to varying $CO_2$ in our simulations. However, a meta-analysis of plant responses to elevated $CO_2$ found that plants in low P environments have lower biomass growth than those in high P (Jiang et al., 2020). This suggests that higher C investment in P acquisition strategies under elevated $CO_2$ has only a limited role in alleviating P stress, particularly when N required for these strategies is limiting. Finally, regarding vegetation biomass and GPP, the co2var scenario was seen to have a much larger effect than tempvar (Figs. A5-A6, items h and i).

**5 Conclusions**

Our inclusion of the P-cycle into a community-developed DGVM has confirmed expectations of the crucial role P plays in vegetation productivity and structure. We have found that P limitation is widespread globally as a co-limiting factor together with N, and as predominant limiting factor in the tropical lowland regions. In addition, the effect of P in limiting global vegetation productivity has been increasing during the last and beginning of this century, becoming more limiting than N

globally after the year 2000. Our process-based model approach allowed us to evaluate which factors were behind this increase, revealing that N limitation decrease was much more driven by temperature changes than increases in N deposition rates. Also, while N limitation changes were predominantly affected by temperature, P changes were mostly affected by $CO_2$ increase. The progressive P limitation may play a significant role in constraining model estimations of $CO_2$ fertilization under future climate change scenarios. The inclusion of the P-cycle in vegetation models is therefore an important step in improving model realism and avoiding productivity overestimations, but further ones may be important too. These include the impact of microbiota and plants in mineral and organic weathering (which can alleviate P limitation in relation to our results), and cycles of other important plant nutrients, such as potassium and calcium (which may alter spatial patterns of biomass and productivity in relation to our results). A more in-depth evaluation of N and P limitation trends over the last century and with future projections using a suite of models which include the N and P cycle and other P-related processes not included here, would be invaluable to confirm if progressive P limitation is under way on a global scale.

**6 Appendix A**

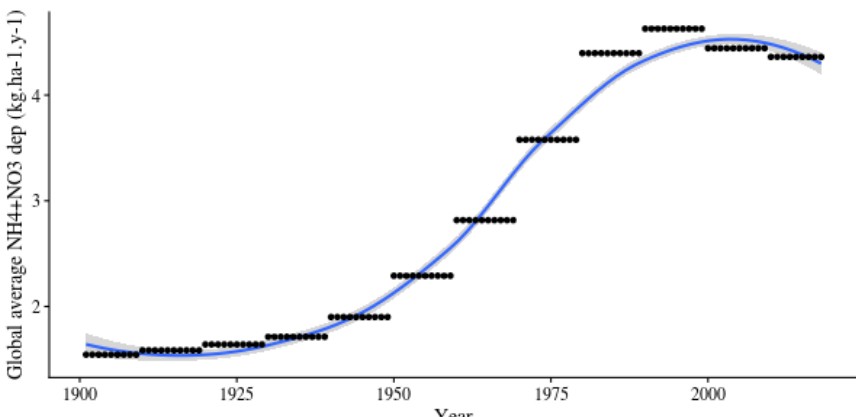

**Figure A1. Yearly trend of global N deposition for the LPJ-GUESS runs, based on the ACCMIP dataset (Lamarque *et al.*, 2013).**

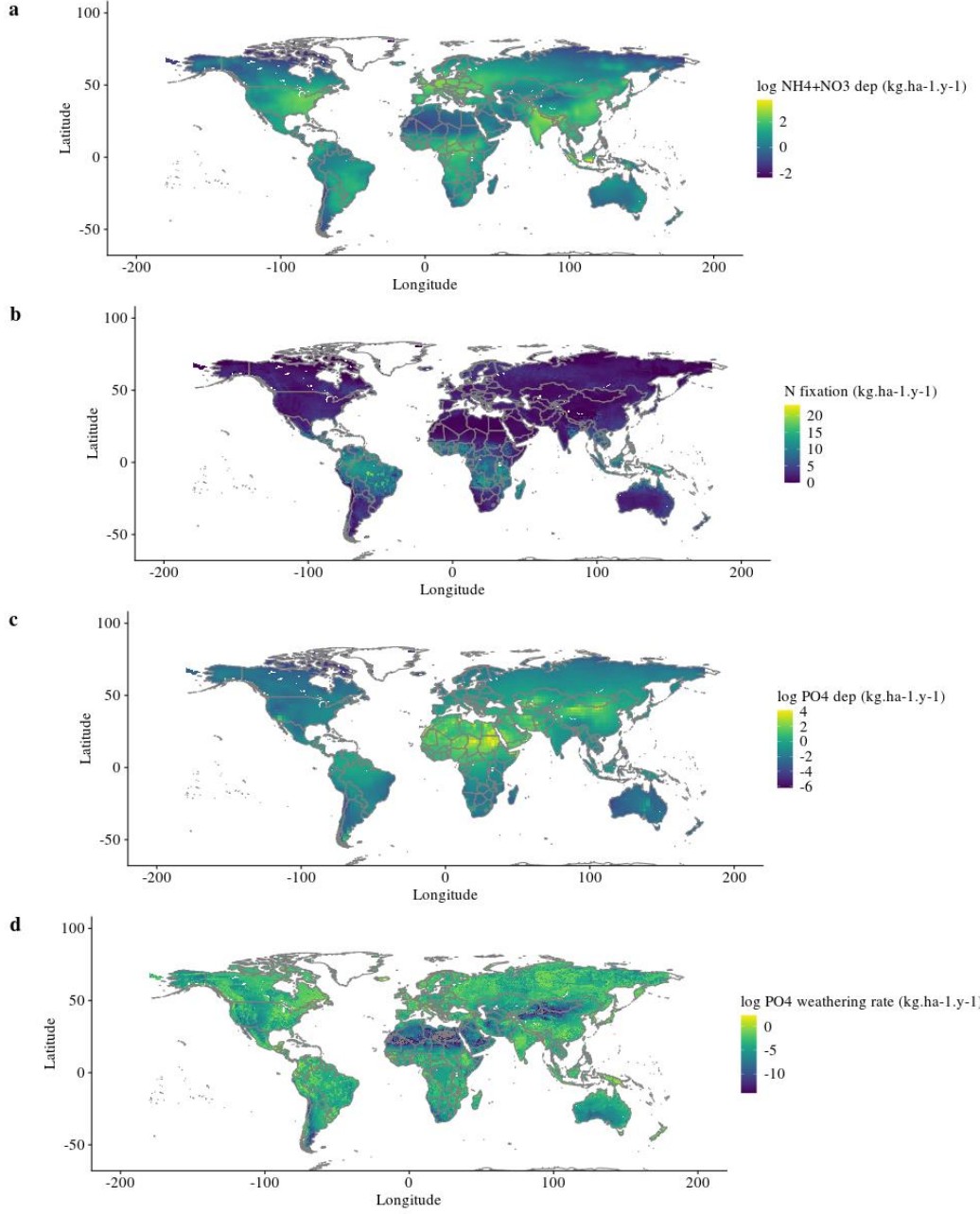

**Figure A2. Global inputs (gridcell averages for the period 1901-2018) of N and P. (a) Inorganic N deposition, (b) N fixation, (c) Inorganic P deposition, (d) P weathering rate.**

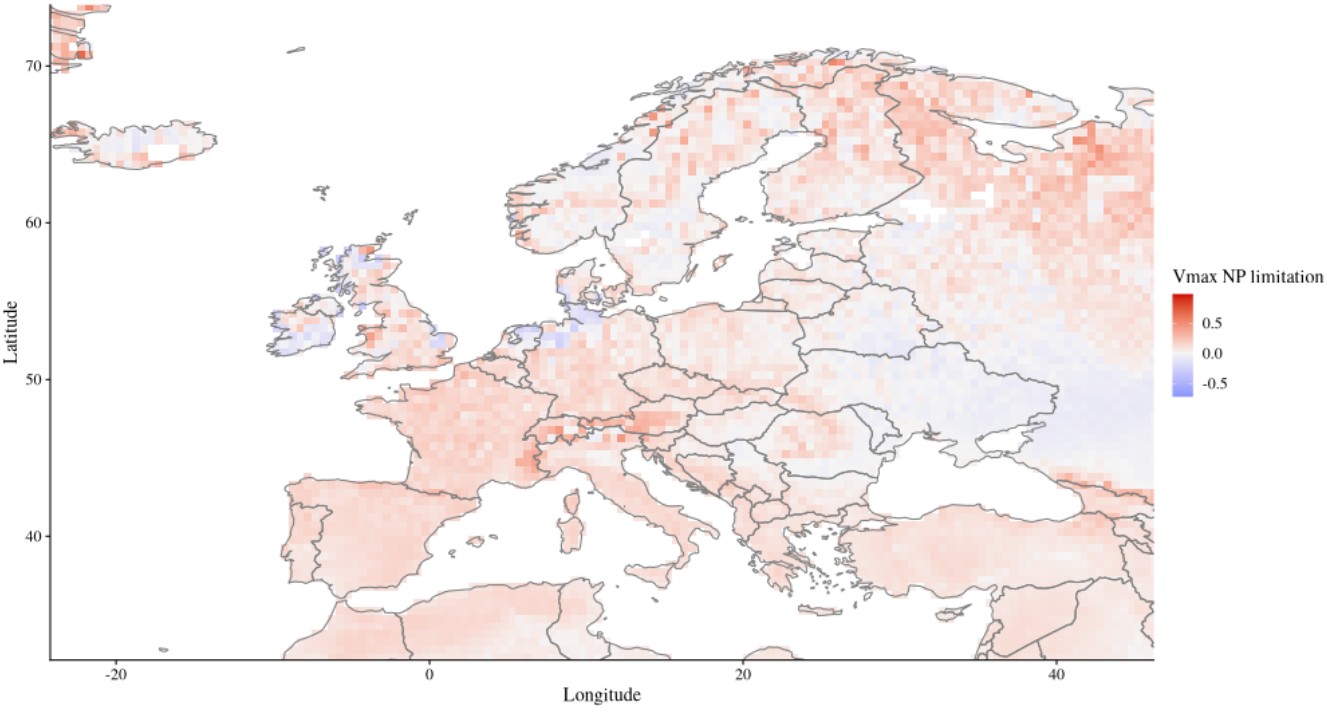


**Figure A3. Map of predominant N or P limitation for Europe, with positive values being predominantly N limited, and negative values predominantly P limited. Values for each gridcell are averages from 2005 - 2018.**

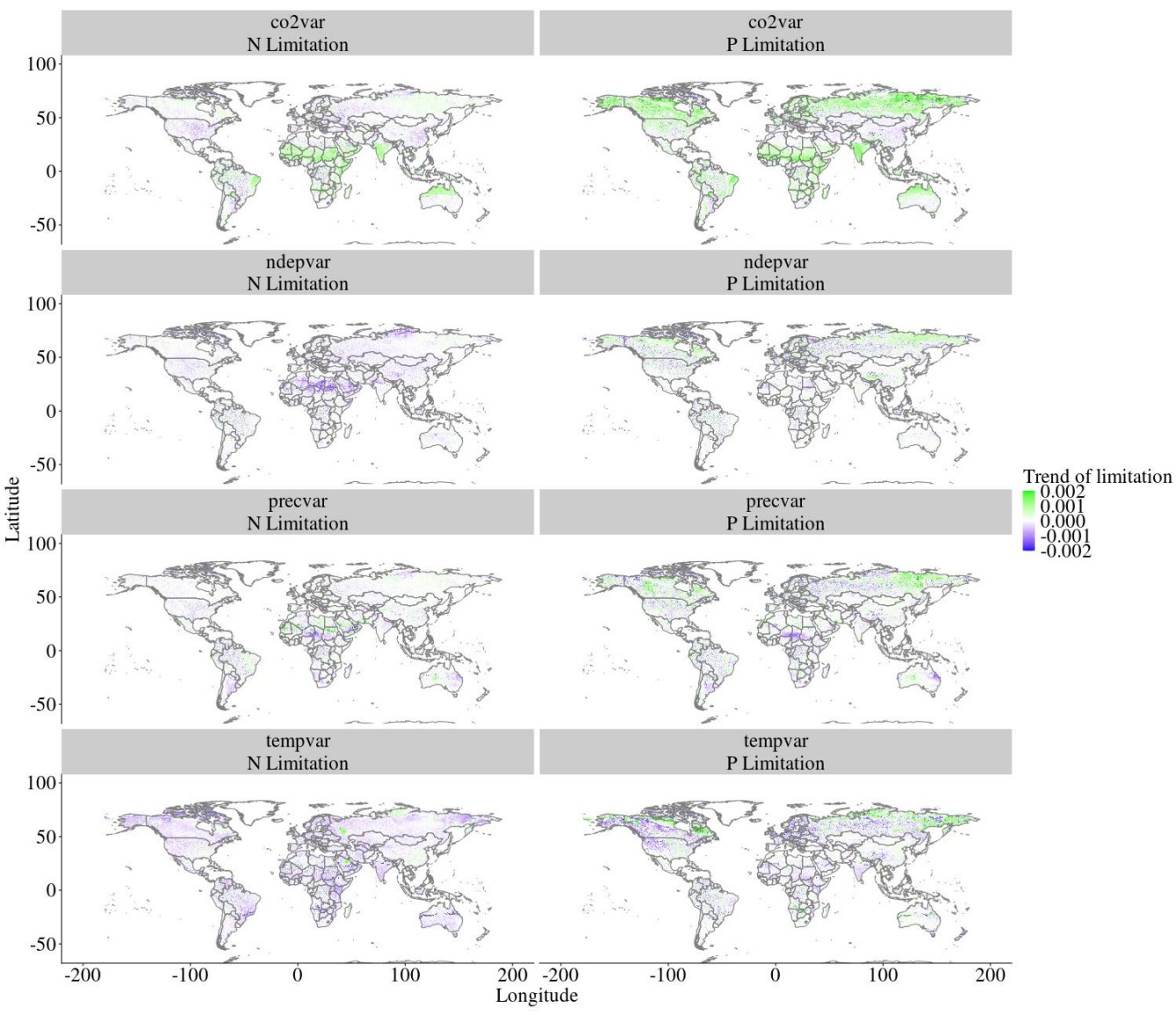

**Figure A4. Spatial trends of N and P limitation change between 1901-2018 for the factorial experiment scenarios. Values are slopes of trends, with positive values (green) indicating increasing limitation, and negative values (purple) indicate decreasing limitation.**

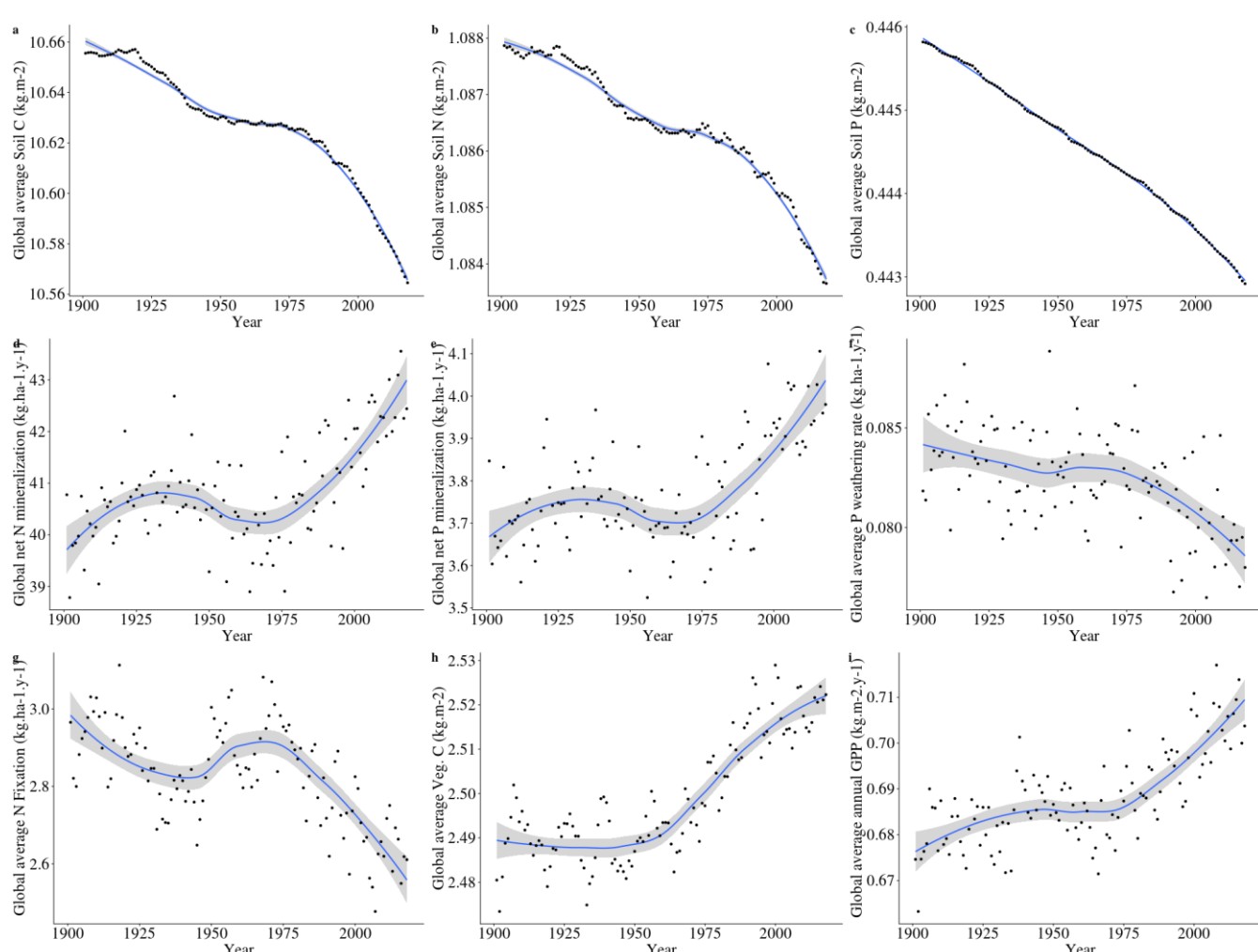

**Figure A5. Soil C, N and P fluxes for the tempvar scenario from 1901 to 2018, with points representing global average values for each year, the lines representing trends (loess), and the gray area the confidence interval (0.95).**


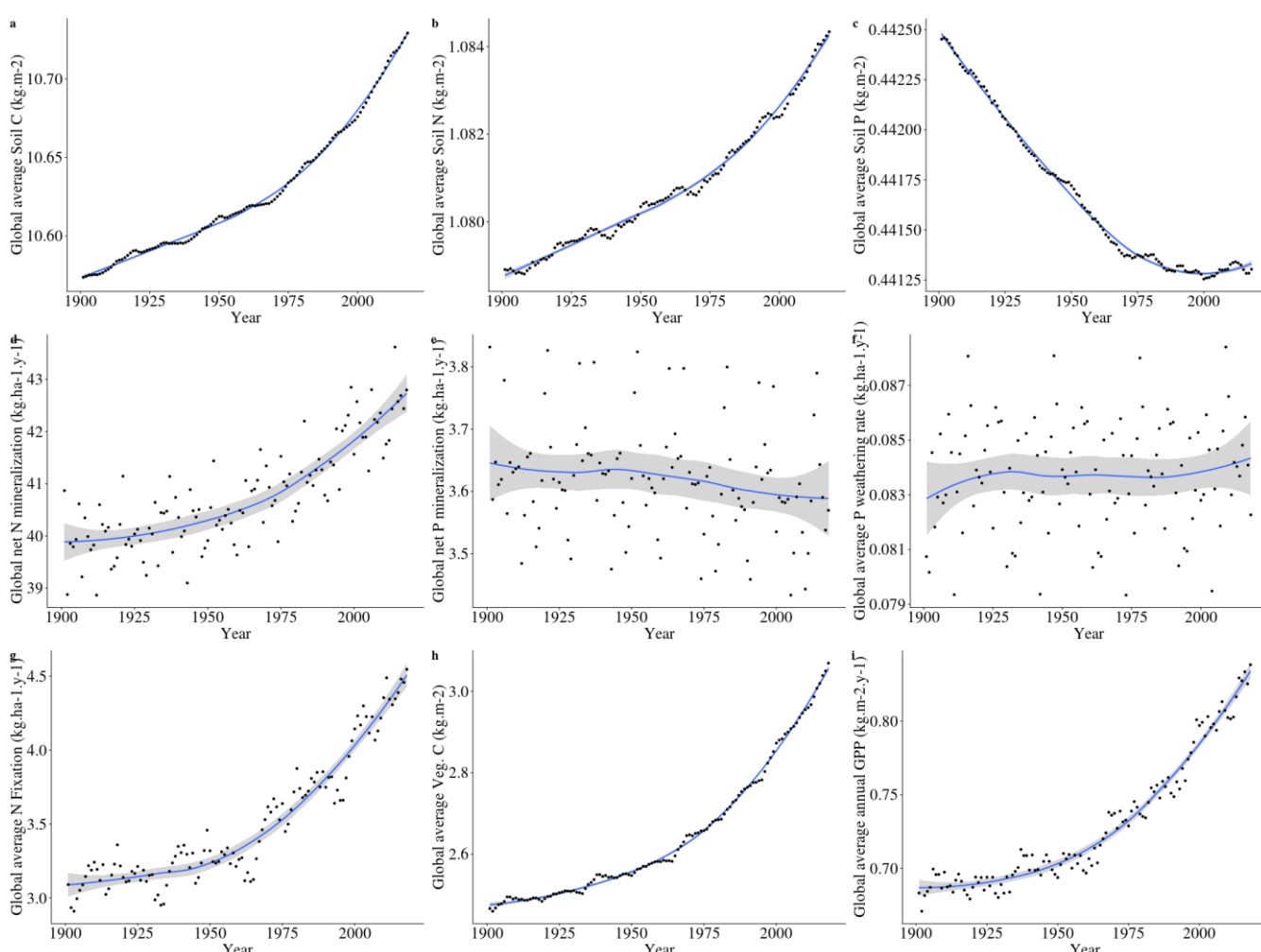

**Figure A6. Soil C, N and P fluxes for the co2var scenario from 1901 to 2018, with points representing global average values for each year, the lines representing trends (loess), and the gray area the confidence interval (0.95).**



## 6.1 Soil phosphorus parameters

| Soil texture (order) | $k_{plab}$ (gP m$^{-2}$) | $S_{pmax}$ (gP m$^{-2}$) |
|---|---|---|
| Ice (Inceptisol/Gelisol) | 65 | 77 |
| Coarse (Inceptisol) | 65 | 77 |
| Medium (Entisol - Alfisol) | 75 | 134 |
| Fine (Inceptisol) | 65 | 77 |
| Medium-coarse (Oxisol/Ultisol) | 10 | 145 |
| Fine-coarse (Ultisol) | 64 | 133 |
| Fine-medium (Aridisol) | 78 | 80 |
| Fine-medium-coarse (Inceptisol) | 65 | 77 |
| Organic (Histosol) | 65 | 77 |
| Vertisols | 32 | 32 |

**Table A1. Parameters for equilibrium between sorbed and labile P ($k_{plab}$) and the maximum amount of sorbed P ($S_{pmax}$) parameters for each of the LPJ-GUESS soil types. Based on Wang et al., (2010).**

**6.2 Soil phosphorus dynamics**

The equilibrium relationship between labile P (Plab) and sorbed P (Psorb) is determined by the Langmuir equation, based on Wang *et al.*, (2010), being defined as:

$$\frac{dPsorb}{dt} = \frac{kplab\ Spmax}{(kplab + Plab)^2}\frac{dPlab}{dt} \quad (Eq.A1)$$


Where $k_{plab}$ and $S_{pmax}$ are soil dependent parameters, and dPsorb/dt and dPlab/dt are described as positive or negative fluxes. The flux from the sorbed pool to the strongly sorbed pool (dPssorb/dt) is also based on the same previous reference, and defined as:

$\frac{dPssorb}{dt} = \mu sorb\ Psorb - \mu ssorb\ Pssorb$  (Eq.A2)

Where $\mu sorb$ and $\mu ssorb$ are rate constants and both equal 0.0067 y-1.

### 6.3 Plant P stoichiometry

Leaf average C:P ratio, which determines plant P demand, is defined using a global tradeoff equation from the TRY database (Kattge *et al.*, 2020), being defined as:

$$log(C:P_{leaf}) = 8.633 + log(SLA) \cdot -0.809$$  (Eq.A3)

where SLA is the individual´s specific leaf area, as defined from the parameter file for each plant functional type. From this average, minimum and maximum C:P values are defined as +/- 39% of the average C:P (same approach as for C:N, Smith et al. 2014). If uptaken P (C:Pactual) is not enough satisfy the average C:P ratio, the plant is under stress at a factor of phosphorus stress pstress = C:Pactual/C:Paverage. The minimum of pstress, nstress (same calculation with N) and water stress will be multiplied by the leaf:fine root allocation ratio and determine how much more biomass will be allocated to

roots than leaves.

The C:P ratio of other tissues is calculated as a function of the leaf values, following the same proportions as those of N defined in the CN version of GUESS. C:P ratios of roots are 1.16 times those of leaves, and C:P ratios of sapwood is 6.9 times those of leaves.

### 6.4 P effects on photosynthesis


The maximum Carboxylation rate ($Vc_{max}$) dependent of leaf P content is defined as:

$$Vc_{max,P} = 11.10 + P_{active} \cdot 0.353$$  (Eq.A4)

Where $Vc_{max,P}$ is in kgC m$^{-2}$ d$^{-1}$ and $P_{active}$ is in kgP m$^{-2}$ and the phosphorus fraction related to photosynthesis, being defined as:

$$P_{active} = P_{leaf\ total} - 3.145\ 10^{-4} \cdot C_{leaf\ total}$$  (Eq.A5)

Both $Vc_{max,P}$ and $P_{active}$ equations are based on Hidaka & Kitayama, (2013).

### 6.5 P weathering

Phosphorus weathering is based on the chemical weathering model of Hartmann & Moosdorf, (2011) and Hartmann *et al.*, (2014). This empirical model is driven by the Global Lithological map database (GLiM, (Hartmann & Moosdorf, 2012)), which divides the surface into classes from which the parameters for the model are derived. For each gridcell and time step, P weathering is then determined by:

$$F_{PW} = F_{CW,i}(lithology, runoff) \cdot F_T(temperature) \cdot F_{s,i}(lithology) \text{ (Eq.A6)}$$

Here, $F_{CW,i}$ is the chemical weathering rate in t km$^{-2}$ y$^{-1}$ for the lithological class *i*, defined by:

$$F_{CW,i} = (b_{carbonate} + b_{silicate})_i \cdot p_i \cdot q \text{ (Eq.A7)}$$

Being $b_{carbonate}$ and $b_{silicate}$ are the chemical weathering parameters (Hartmann *et al.*, 2014, Table A1-1) for carbonate and silicate respectively, $p_i$ is the phosphorus content (Hartmann *et al.*, 2014, Table A1-2) of the lithological class *i*, and *q* is the runoff in mm years, which is included prognostically in the model for each simulated patch.

Also, FT is the temperature effect on weathering, calculated by:

$$F_T = e^{\left(\frac{-E_{a,i}}{R} \cdot \left(\frac{1}{T} - \frac{1}{T_0}\right)\right)} \text{ (Eq.A8)}$$

Where $E_{a,i}$ is the activation energy for the lithological class *i*, *R* the gas constant, *T* the soil temperature in Kelvin, calculated prognostically in the model daily for each gridcell, and $T_0$ the average reference temperature of Japan (284.15 K).

Finally, to account for some conditions where weatherable material is isolated from the hydrological processes by thick, chemically depleted soils or other surface layers such as wetlands or laterites, a soil shielding factor was included for some lithologies (Hartmann *et al.*, 2014). $F_s$ is then 0.1 for the lithologies where soil shielding has a significant role, and 1 otherwise.

| | P limited % |
|---|---|
| Area | 45.9 |
| GPP | 57.7 |
| NPP | 62.6 |
| Veg Biomass | 68.4 |
| Total Organic C | 34.2 |
| Total C | 44.1 |

**Table A2. Percentage of several ecosystem measures in P limited regions (defined as negative $Vc_{max,NPlim}$, calculated using Eq. 3, main Text, and averaged from 2005 to 2018).**

**7 Code availability**

The current version of LPJ-GUESS-CNP model is available from https://github.com/mateusdp/LPJ-GUESS-NTD/tree/phosphorus-cf-walker under the Mozilla Public License 2.0. The exact version of the model used to produce the results used in this paper is archived on Zenodo (DOI: 10.5281/zenodo.13471785), as are input data (10.5281/zenodo.13472421 for the standard simulation runs, and 10.5281/zenodo.13594436 for the factorial simulation runs) and scripts to run the model and produce the plots for all the simulations presented in this paper (DOI:
10.5281/zenodo.13385547).

**8 Author contributions**

Mateus Dantas de Paula: design of the research; performance of the research; data analysis and interpretation; writing the manuscript

Matthew Forrest: design of the research; performance of the research; data analysis and interpretation; writing the
manuscript

David Wårlind: design of the research; performance of the research; data analysis and interpretation; writing the manuscript

João Paulo Darela Filho: performance of the research; data analysis and interpretation; writing the manuscript

Katrin Fleischer: performance of the research; data analysis and interpretation; writing the manuscript

Anja Rammig: performance of the research; data analysis and interpretation; writing the manuscript
Thomas Hickler: design of the research; performance of the research; data analysis and interpretation; writing the manuscript

**9 Competing interests**

The contact author has declared that none of the authors has any competing interests.

**10 Acknowledgements**

We would like to thank Natalie Mahowald, Jens Hartmann and Giovanni Iadarola for support in the development of this
study.

Mateus Dantas de Paula acknowledges financial support from the DFG funded research unit RESPECT (grant no. FOR2730).

David Wårlind acknowledges financial support from the Strategic Research Area MERGE (Modeling the Regional and Global Earth System - www.merge.lu.se) and the Swedish Research Council (grant no. 2020-05051).

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
