# Peer review of "Including the Phosphorus cycle into the LPJ-GUESS Dynamic Global Vegetation Model (v4.1, r10994) – Global patterns and temporal trends of N and P primary production limitation."

_EGUsphere, 2024_

## Author Response (AR1)

*Note: The line numbering of the reviewer's comments refer to the unrevised manuscript version (P_Cycle_GUESS_Dantas_de_Paula_GMD_Format_Main_Text_min_rev1.1) and the line numbering of the replies refer to the revised manuscript version with tracked changes (P_Cycle_GUESS_Dantas_de_Paula_GMD_Format_Main_Text_review_tracked_012025).*

**REVIEWER 1**

This manuscript clearly explains the method and performance of integrating the phosphorus cycle into the LPJ-GUESS Dynamic Global Vegetation Model. The writing is logically structured, and the model development process is well-reasoned. The authors also provide a thorough discussion of the results. I have only a few detailed comments, which are outlined below.

Introduction

The introduction does a good job in establishing the need for incorporating the phosphorus cycle into dynamic vegetation models, highlighting its importance as a plant growth constraint. However, it would benefit from a more thorough discussion of past challenges in phosphorus modeling and recent advancements, especially regarding prior efforts in Earth system models, where phosphorus cycling, though less common than nitrogen, has been integrated in models like ORCHIDEE-CNP and CASA-CNP. Clarifying research gaps and this study's unique contributions compared to these models would enhance its impact. Additionally, providing context for the four scientific questions would help readers understand their relevance, as they currently appear somewhat abruptly.

Methodology

In Figure 1, consider adding some equations (or other elements) to illustrate the principles behind phosphorus limitation, such as how soil phosphorus cycling regulates available phosphorus levels and how soil phosphorus uptake limits plant photosynthesis. This would help readers better understand the model's configuration, especially as this is a central focus of the study. Additionally, is the leaching path missing from the figure?

I recommend moving the content related to P soil processes from the appendix into the main text, particularly sections on P sorption and P weathering. For readers less familiar with these processes, equations can facilitate understanding.

Integrating phosphorus limitation into the model is indeed a significant advancement. However, we know that plant growth may also be limited by other elements, such as potassium or calcium. Therefore, there is still the potential to overestimate plant growth in this model (at least from a nutrient supply perspective). I suggest adding a few sentences in the discussion to address this limitation.

Results

The results are well-documented.

Discussion

The discussion section provides a thorough evaluation of the model's performance and areas for improvement. I suggest more discussion of potential limitations. For example, the model lacks certain processes in the phosphorus cycle, such as the impact of plants and microbes on weathering, and there is uncertainty in data sources, as phosphorus content in rock is typically based on sample averages that may not fully represent the mineral composition across all regions. Highlighting these limitations would inspire future research to address these aspects.

**REPLY to REVIEWER 1**

*Many thanks for your comment to the manuscript, the suggestions have been duly appreciated:*

*Introduction*

*The paragraph starting in line 62 was extended to include additional arguments in favour of including the phosphorus cycle in DGVMs, as well as a larger list of existing models such as ORCHIDEE and CASA-CNP which include this cycle. In addition, the research gaps in these previous studies were included, namely a global spatial and temporal analysis of N and P limitation, also through a factorial analysis, which in addition provide more context for the four scientific questions which were presented.*

*Methodology*

*References to the equations illustrated by Figure 1 were added, as well as further details on how P in soil may affect decomposition rates. Hopefully this increases the understanding of the model´s configuration. The leaching path that was absent was included.*

*Due to text size constraints, the equations were kept in the appendix. It is believed this would not hinder access to this information, since in the journal´s format the Appendix are embedded into the main document, and not a separate one.*

*Model limitations in relation to absent processes were added in the manuscript´s final paragraph.*

*Discussion*

*A few lines on the uncertainty of P release estimations based on the Chemical Weathering Model and due to the limited amount of sampling was added to the last paragraph of section 4.2.*

**REVIEWER 2**

My recommendation is minor revision or rather technical corrections. The manuscript is well written and structured, and most importantly, the transparency of the model assumptions and evaluation. The aim with the extended LPJ-GUESS-CN with the P cycle was to addresses the following research questions (1) Does including the P-cycle improve model agreement to biomass and gross primary production (GPP) observations? (2) What drives N and P limitation across climate zones? (3) How has that changed during the 20th and early 21st century? and (4) Which environmental factors are more relevant for N and P limitation change? Questions 1-3 are clearly answered, and the answers form also the major contribution of the novelty value of the manuscript. Concerning research question (4), I recommend a specification which environmental factors are studied or in mind. Do you mean drivers of plant growth such as N deposition, temperature, precipitation and ambient CO2-concentrations or do you have other factors in mind?

Concerning the P cycle concept, I have one question or minor concern considering the effect of fire on P cycle. L 124 It says "we consider burnt P to be completely retained in the soil". Do you mean burnt P in the soil or burnt P total plant? Earlier, the authors mention that in the tropical region, most of P is found in the biomass. Hence, I rather expected an addition of both burnt fine and burnt coarse woody debris. Adding specific pools and/or cohorts for burnt debris, would also allow to include the impact of burnt debris or charcoal on decomposition and sorption processes. I have the impression that disturbance by fire was not applied in this study, and thus the results shown are independent of the fire concept. Yet, fires are expected to become more frequent and intensive, and for future use of LPJ-GUESS-CPN, it maybe of importance.

Detailed comments

L 65 Cramer, W., Bondeau, A., Woodward, F.I., Prentice, I.C., Betts, R.A., Brovkin, V., Cox, P.M., Fisher, V., Foley, J., Friend, A.D., Kucharik, C., Lomas, M.R., Ramankutty, N., Sitch, S., Smith, B., White, A., Young-Molling, C., 2001. Global response of terrestrial ecosystem structure and function to CO2 and climate change: results from six dynamic global vegetation models. Global Change Biology 7, 357e373. Is a relevant reference here.

L 69 Wrong reference for He et al. 2021. Change to He et al. 2021a and add He H., P-E Jansson, A.I. Gärdenäs. 2021. CoupModel (v6.0): an ecosystem model for coupled phosphorus, nitrogen and carbon dynamics – evaluated against empirical data from a climatic and fertility gradient in Sweden. Geoscientific Model Development 14, 735–761, doi.org/10.5194/gmd-14-735-2021.

The later references to He et al. 2021 such be changed to He et al. 2021b

2.5 Driving data

L184 Add the ranges in N and P deposition, so that the reader get an impression of magnitude.

Other driving data?

Table 1 Heading, denote CN and CNP-versions same as in text.

L281 Please clarify is water runoff considered important for transport and losses of weathered P or for weathering process?

L356 Here He et al. 2021a is relevant. Please clarify why plant root-symbiosis maybe more important for plant P than for plant N acquisition. Slight overlap with paragraph L425.

Figures A5 and A6 incomplete figure legend; dots, lines and area denote …

**REPLY to REVIEWER 2**

*Thank you for your input, the suggestions have been appreciated and incorporated to the manuscript as follows:*

*Introduction*

*L87: I have added the factor list (CO2, N deposition, precipitation and temperature) that were investigated into the question (4) in order clarify exactly which were evaluated.*

*Fire*

*This was well noted many thanks. The GLOBFIRM module which was used in this study considers the burning of both living plant tissue and litter, which considers fine and coarse woody debris. In all P is completely retained to the soil after burning. I have added this clarification to line 137. The inclusion of specific pools for burnt debris and charcoal is a fascinating concept; however these are not included in the simple fire model that was used. We did not investigate in depth the effect of fire into the P cycle and how it then cascades into the complete ecosystem, although it was included, but using newer fire implementations of LPJ-GUESS, such as SPITFIRE (Thonicke et al., 2010) would definitely allow further analysis of these impacts.*

*L66: thanks for the suggestion, the reference to Cramer et al. has been added.*

*L76: Thank you, the reference to the CoupModel was added as He et al. 2021b.*

*L197: The ranges of N and P deposition have been added, thanks.*

*Table 1. The model version names have been updated.*

*L306: Thank you for noting this, both weathering and losses (leaching) of P are dependent on runoff, this has been corrected in the text.*

*L380: The phrases regarding symbiotic nutrient acquisition were removed from this paragraph, since in the paragraph of L425 already discusses this. Thanks for noting*

*Figures A5 and A6: Thanks for noting, the description now reads: "points representing global average values for each year, the lines representing trends (loess), and the gray area the confidence interval (0.95)"*

*References*

*Thonicke K, Spessa A, Prentice IC, Harrison SP, Dong L, Carmona-Moreno C. 2010. The influence of vegetation, fire spread and fire behaviour on biomass burning and trace gas emissions: Results from a process-based model. Biogeosciences 7: 1991–2011.*

**REVIEWER 3**

In this study, Dantas de Paula et al presented the phosphorus cycle into LPJ Guess model. Given the emerging importance in P on regulating the C cycle, the N cycle, and vice versa, several DGVMs are incorporating P into its model structure, this among one of its kind. The authors discuss mainly on the GPP and the stocks' part and conducted a sensitivity analysis for climatic and edaphic drivers. The CNP model fits with the empirical stock data, showed expected P limitation geological distributions. Factor experiments suggest N being more temperature dependent and P to CO2 changes. These results seem sound and logic. Overall, the paper is well written and clear, thus I would recommend publishing at GMD. The other reviewers have made many comments, here I would emphasize the following which I believe can further improve the manuscript to advance the P incorporation in ecosystem models.

In my view, the overall P model development is mostly centred on two questions, Frist, How N and P limitations are determined, and co-regulations/co-limitations are designed, further how the interaction with C, e.g. GPP calculation is structure and parameterised. These two questions essentially determine the output of the research questions the authors want to address here and beyond. I would rather suggest the authors compare the design of those in LPJ with existing models (although I do see many processes follows existing approaches), and how the difference design or representations make a difference or not? To me this can be an important implication to the community of this model development here.

I would like to raise the importance of parameterisation, since the outputs e.g. table 1, is essentially a product of parameterisation, thus it is crucial to present and discuss which is the crucial parameter, how those parameters are determined, how it compares with empirical process studies. For instance, the discussion of soil organic P and labile P stocks at Line 340 to me is again a parameterisation issue. A major drawback for P model developments is the scarcity of data for parameterize the major soil-plant fluxes, for instance, the weathering/deposition flux, particularly the sorption-desorption dynamics in the soil is the key in regulating the root-fungi-soil continuum in P availability. Discussion of the parameters would also add the implications of current modeling study to further measurements to improve our future understanding the coupled C-N-P cycle.

What is the explanation for the increase of GPP with factoring P cycle (e.g Line 236), and why is so? Would this relate to feedback of the SOM decomposition? What processes in the model is suggesting this? Is this seeing from other models?

I would also suggest discussing further the drawbacks of current model configuration. e.g. The vegetation-soil P cycle is quite tightly coupled, meaning the mineralization and/or root- fungi uptake is the dominant flux in many cases. Given the soil components in LPJ is simplified, e.g. Line 121-124. How does that influence the P cycle? What's the limitations of that.

Finally, another way to present the stocks and fluxes are present the stoichiometry, and compare with the empirical data.

The paper is overall well written, some minor suggestions below:

Line 62 what is large time and spatial scales, do you mean P limitation occurred in the geological scale? But DGVMs normally do not run in those scales, so what better be specific here.

Line 112 Precisely the reference should be Wang et al 2007 GBC?

Line 237 why is that, also see above

Table 1 be specific about the "Soil", in the "Litter+Soil" term.

Line 246 Table 2, the two versions are rather the similar if you consider the uncertainties in your reference dataset, a slight change in r2 and RMSE, is surely non-significant particularly when you compare the global averages. To me it would be more logical to present this as a benchmark, then discuss the model setup and parameterisation, rather than arguing an improved model performance..

Fig 4, legends a bit too small to see

Fig 5, the same what is soil P here?

**REPLY to REVIEWER 3**

*Thank you very much for this insightful review.*

*General*

*Indeed the many phosphorus cycle implementations we present here have already been used in other models, but such a model intercomparison would be invaluable to understand how modelling approaches regarding parametrization or other differences impact results and allow consensus. However, to our current knowledge this has been very few global modelling the effects of the P-cycle in vegetation (Wang et al., 2010), and a model comparison on a local scale has already been conducted (Fleischer et al., 2019). It is expected that with further P-cycle model development global comparison studies can be carried out.*

*The topic of parametrization is very important, as the reviewer pointed out. In LPJ-GUESS-CNP phosphorus related parameters are taken from the literature, and not calibrated. Michaelis-Menten kinetics parameters for P uptake were missing and now are added to the text. Understanding the relevance of each parameter would also be very useful in a sensitivity analysis, but one which unfortunately was not our focus here.*

*Here I assume the reviewer means decrease of GPP with the inclusion of the P-cycle? If this is the case, discussion of this pattern can be found in sections 4.1 and 4.2.*

*Regarding P-cycle coupling, this was well noted. The P-cycle implementation used in this study already had a change in which organic P is not leached but retained and mineralized, in order to reflect the tighter cycling of phosphorus. This was corrected in Line 167.*

*L62: Thank you for noting this, here the temporal scales referred here are the decadal to century scales normally used in DGVMs, as opposed to the temporal scales of field experiments, normally a few years. A clarification was added.*

*L125: Well noted, the reference was updated to include the 2010 and 2007 publications.*

*L237: Please see also above, detailed explanations in sections 4.1 and 4.2. An extra reference was added regarding the increase of P limitation outside tropical regions (L434).*

*Table 1. Thank you for the observation, a more precise description of litter and soil matter pools was added to the Table 1 legend.*

*Table 2. It is true that from a global perspective the R2 and RMSE differences are small, but these differences occur predominantly where we would expect, in the P-limited tropical regions (particularly for biomass).*

*Figure 4. Thank you for noting, the figure size was increased.*

*Figure 5: Similar description of Table 1 of soil and litter was added here, thank you for noting.*